# Inhibitors of the small membrane (M) protein viroporin prevent Zika virus infection

Emma Brown[1,2‡], Gemma Swinscoe[1,2,3], Daniella A Lefteri[2§], Ravi Singh[1,4#], Amy Moran[1,2], Rebecca F Thompson[1,3], Daniel Maskell[1,3], Hannah Beaumont[1,2], Matthew J Bentham[1,2], Claire Donald[5], Alain Kohl[5], Andrew Macdonald[1,3], Neil Ranson[1,3], Richard Foster[1,4†], Clive S McKimmie[2†], Antreas C Kalli[1,6†], Stephen Griffin[1,2*]

[1]Astbury Centre for Structural and Molecular Biology, University of Leeds, Leeds, United Kingdom; [2]Leeds Institute of Medical Research, School of Medicine, Faculty of Medicine and Health, University of Leeds, St James' University Hospital, Leeds, United Kingdom; [3]School of Molecular and Cellular Biology, Faculty of Biological Sciences, University of Leeds, Leeds, United Kingdom; [4]School of Chemistry, Faculty of Maths and Physical Sciences, University of Leeds, Leeds, United Kingdom; [5]MRC and University of Glasgow Centre for Virus Research, Sir Michael Stoker Building, Glasgow, United Kingdom; [6]Leeds Institute for Cardiovascular and Metabolic Medicine, Faculty of Medicine and Health, University of Leeds, Leeds, United Kingdom

*For correspondence:
s.d.c.griffin@leeds.ac.uk

†These authors contributed equally to this work

Present address: ‡Department for Health and Social Care (DHSC), Leeds, United Kingdom; §MRC and University of Glasgow Centre for Virus Research, Glasgow, United Kingdom; #Department of Chemistry, Molecular Sciences Research Hub, London, United Kingdom

Competing interest: The authors declare that no competing interests exist.

**Abstract** *Flaviviruses*, including *Zika virus* (ZIKV), are a significant global health concern, yet no licensed antivirals exist to treat disease. The small membrane (M) protein plays well-defined roles during viral egress and remains within virion membranes following release and maturation. However, it is unclear whether M plays a functional role in this setting. Here, we show that M forms oligomeric membrane-permeabilising channels in vitro, with increased activity at acidic pH and sensitivity to the prototypic channel-blocker, rimantadine. Accordingly, rimantadine blocked an early stage of ZIKV cell culture infection. Structure-based channel models, comprising hexameric arrangements of two *trans*-membrane domain protomers were shown to comprise more stable assemblages than other oligomers using molecular dynamics simulations. Models contained a predicted lumenal rimantadine-binding site, as well as a second druggable target region on the membrane-exposed periphery. In silico screening enriched for repurposed drugs/compounds predicted to bind to either one site or the other. Hits displayed superior potency in vitro and in cell culture compared with rimantadine, with efficacy demonstrably linked to virion-resident channels. Finally, rimantadine effectively blocked ZIKV viraemia in preclinical models, supporting that M constitutes a physiologically relevant target. This could be explored by repurposing rimantadine, or development of new M-targeted therapies.

## Editor's evaluation

This study presents a valuable finding on the viroporin activity of the ZIKV M protein. The evidence supporting the claims of the authors is solid. The work will be of interest since M protein could be a relevant target for the development of new therapies.

 

## Introduction

*Zika virus* (ZIKV) was first isolated in the eponymous Ugandan forest in 1947 and is a mosquito-borne arbovirus (<u>arthropod-borne</u>), classified within the *Flavivirus* genus of the *Flaviviridae* (*Dick, 1952*; *Dick et al., 1952*). Historically, ZIKV caused infrequent and relatively mild infections in humans with symptoms comprising fever, rash, headaches, joint/muscle pain, and conjunctivitis (*Simpson, 1964*). However, once confined to Africa and Southeast Asia, a large outbreak in 2007 on Yap Island (Micronesia), followed by further outbreaks in French Polynesia, New Caledonia, and the Cook Islands in 2013/14, signified spread of the virus across the Pacific. The virus then arrived in Brazil in 2015, where an epidemic affecting >1 M people ensued. This resulted in worldwide spread of the virus involving more than 80 countries and signified emergent ZIKV as a global health concern (*Lazear and Diamond, 2016*).

Worryingly, the outbreaks in Yap and thereafter encompassed considerably more serious symptomology, which has been linked to evolution of New World ZIKV strains (*Gong et al., 2017*). This included more serious and frequent neurological complications, increased incidence of Guillian–Barre syndrome, and a devastating epidemic of microcephaly in new-borns where maternal infection had occurred during the first trimester of pregnancy (*Rasmussen and Jamieson, 2020*; *Rasmussen et al., 2016*; *Rasmussen et al., 2017*). Eventually, the outbreak was contained by surveillance combined with control of insect vectors, predominantly *Aedes aegypti* and *Aedes albopictus* mosquitos (*Peterson et al., 2016*). More than 7 years later, a ZIKV vaccine remains elusive, although several are in trials (reviewed in *Poland et al., 2018*). Similarly, no licensed antiviral therapies exist to treat ZIKV infection, yet repurposing of nucleotide analogue inhibitors against the NS5 protein shows promise (*Mumtaz et al., 2017*; *Masmejan et al., 2018*; *Lim et al., 2020*).

ZIKV, like other *Flaviviruses*, possesses a positive sense single-stranded RNA genome over 10 kb in length, capped at the 5′ end by a methylated dinucleotide (m$^7$GpppAmp) (*Garcia-Blanco et al., 2016*; *Ray et al., 2006*). The RNA is translated into a polyprotein that is subsequently cleaved by host and viral proteases to yield 10 mature gene products (*Assenberg et al., 2009*); these are spatially organised into structural virion components at the N-terminus, and non-structural replicase proteins at the C-terminus. The mature enveloped virion comprises the viral RNA, the capsid (C) protein and two membrane proteins: envelope (E) and the small membrane (M) protein (*Prasad et al., 2017*; *Tan et al., 2020*). Virions undergo maturation as they traffic through the cell during egress, with M and the precursor (pr) peptide acting as chaperones for the E protein (*Yu et al., 2008*). Maturation occurs upon furin-mediated cleavage of pr-M in the Golgi (*Stadler et al., 1997*; *Sevvana et al., 2018*), although this has variable efficiency (*Dowd et al., 2014*; *Junjhon et al., 2008*). The cleaved pr peptide remains associated with E, preventing exposure of the fusion loop within acidifying secretory vesicles. Release from the cell into a pH-neutral environment leads to the loss of pr and resultant structural rearrangement of E and M (*Yu et al., 2009*). Whilst E comprises the essential virus attachment and entry protein, the role of M within the mature infectious virion remains unknown (*Sirohi et al., 2016*).

The presence of dimeric M, bound to E dimers (*Sirohi et al., 2016*), within the virion raises the question of whether it might play a role during the virus entry process. Unfortunately, cryo-EM structures of acidified *Flavivirus* particles – a proxy for changes likely to occur during endocytosis – do not contain density attributable to the M protein (*Zhang et al., 2015*). Moreover, E undergoes a shift from a dimeric to a trimeric form thought to mediate membrane fusion (*Bressanelli et al., 2004*; *Stiasny et al., 2004*; *Modis et al., 2005*; *Nayak et al., 2009*; *Zhang et al., 2004*). This presumably disrupts E–M dimeric interactions, thereby allowing M an opportunity to also adopt alternative conformations and explore other oligomeric states.

Interestingly, dengue virus (DENV) M peptides formed membrane channels in vitro (*Premkumar et al., 2005*), which would require the formation of higher-order oligomers. However, electrophysiology studies in *Xenopus* oocytes failed to detect M-mediated proton conductance (*Wong et al., 2011*). Nevertheless, due to its size and hydrophobicity, M has long been speculated to comprise a virus-encoded ion channel, or 'viroporin', typified by the M2 proton channel of influenza A virus (IAV) (*Nieva et al., 2012*; *Royle et al., 2015*; *Scott and Griffin, 2015*). M2 functions to allow protons to pass into the virion core during virus entry, enabling the uncoating of viral RNA, and is the target of the antiviral drugs amantadine and rimantadine, setting clinical precedent for viroporins as drug targets.

Here, we provide evidence that ZIKV M forms rimantadine-sensitive, oligomeric membrane channels whose activity increases at acidic pH. Rimantadine blocks an early step during the ZIKV life cycle,

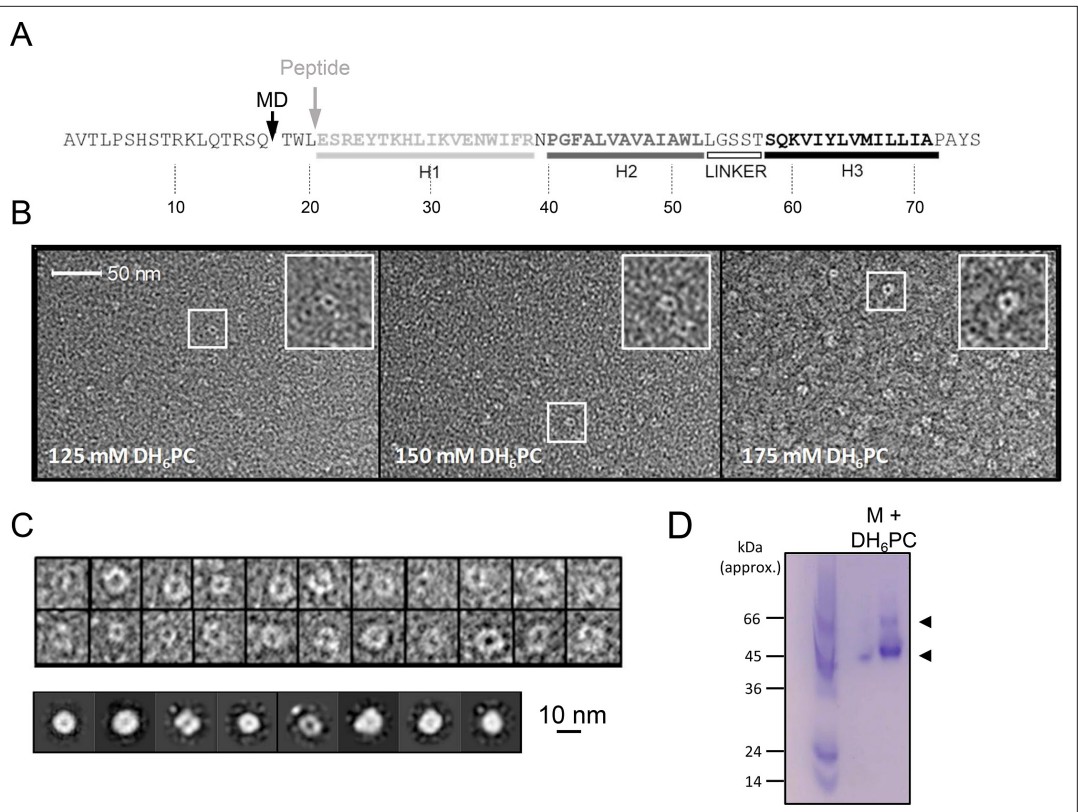

**Figure 1.** in vitro characterisation of M. (**A**) M sequence showing helical regions and peptide truncation (in vitro synthesis (grey arrow) and molecular dynamics (MD) simulations (black arrow)). (**B**) Visualisation of M peptide with increasing concentrations of detergent, stained with uranyl acetate. Fields are representative of multiple images with~9000 particles collected in total across all conditions. Insets show zoomed images of particles with accumulation of stain within central cavity, consistent with channel formation. (**C**) Top panel – examples of particles visualised at 150mM illustrating heterogeneity. 9907 particles from 150mM samples were resolved into two-dimensional (2D) class averages with 25 iterations (bottom panel) yet rotational symmetry could not be determined. (**D**) Native polyacrylamide gel electrophoresis (PAGE) of M peptide (5µg) reconstituted in DH$_6$PC (300mM).

The online version of this article includes the following source data and figure supplement(s) for figure 1:

**Source data 1.** TIFF file of scanned native polyacrylamide gel electrophoresis (PAGE) gel photographed after staining with Coomassie Blue.

**Figure supplement 1.** Example High Performance Liquid Chromatography (HPLC) and mass spectrometry analysis of synthetic M peptides supplied by Alta Biosciences.

**Figure supplement 2.** Predicted membrane protein properties for ZIKV M protein.

consistent with a role for M channels during virus entry. Critically, rimantadine also abrogates viraemia within in vivo ZIKV preclinical models, supporting that M channel activity is a physiologically relevant target. In-depth molecular dynamics (MD) investigations support that M channels are comprised of hexameric complexes, with the channel lumen lined by the C-terminal *trans*-membrane domain (TMD). Ensuing models provided an effective template for in silico high-throughput screening (HTS) of a library of FDA-approved and other biologically active compounds, targeting two discrete sites on the channel complex. Pleasingly, enrichment of the library led to the identification of much-improved blockers of M channel activity, laying the foundation for potential drug repurposing alongside bespoke antiviral development.

# Results

## ZIKV M peptides form channel-like oligomeric complexes within membrane-mimetic environments

M protein resides as a dimeric form resolvable within cryo-EM structures of mature ZIKV, and other *Flavivirus* particles, in close association with E dimers (*Sirohi et al., 2016*). Thus, M would be required to form higher-order oligomers to form a putative channel across the membrane. This might conceivably occur during endocytosis where acidic pH induces reconfiguration of E into a trimeric fusion complex, presumably also liberating M dimers to move within the virion membrane (*Bressanelli et al., 2004*; *Stiasny et al., 2004*; *Modis et al., 2005*; *Nayak et al., 2009*; *Zhang et al., 2004*). Thus, we hypothesised that M peptides should form channel structures within a lipidic environment in the absence of E protein.

A peptide lacking the disordered N-terminal 20 amino acids was synthesised based upon the M protein sequence from a New World ZIKV isolate (PE243) (*Donald et al., 2016*). Peptides therefore comprised the three helical regions of the mature M protein, the second and third of which form TMDs within virion membranes (*Figure 1A*, *Figure 1—figure supplements 1 and 2*). Peptides were reconstituted using several non-ionic detergents used frequently as membrane-mimetics in structural and/or biophysical studies (*OuYang et al., 2013*; *Pielak and Chou, 2010*; *Pielak et al., 2009*; *Schnell and Chou, 2008*), as well as in our previous investigations of viroporin oligomerisation (*Foster et al., 2011*; *StGelais et al., 2009*; *Wetherill et al., 2012*; *Carter et al., 2010*).

To investigate the nature and stoichiometry of M complexes, dihexanoyl-phosphatidylcholine (DH$_6$PC) solubilised M peptides were first immobilised upon carbon grids and negatively stained using uranyl acetate over a range of detergent concentrations. Unfortunately, use of 300 mM detergent prevented the efficient adhesion of peptides, yet visualisation of M complexes at lower concentrations revealed the formation of circular structures, the majority of which were oriented approximately in parallel with the grid plane (*Figure 1B*).

Interestingly, complexes exhibited a significant degree of structural heterogeneity (*Figure 1C*, top panel), although separable class subsets were resolvable amongst >9000 complexes analysed at 150 mM detergent (*Figure 1C*, bottom panel). Unfortunately, the degree of heterogeneity and small size prevented determination of stoichiometry by two-dimensional (2D) rotational averaging. Nevertheless, the absence of obvious aggregation and the formation of, albeit heterogeneous, channel-like structures, supports that M can adopt conformations corresponding to membrane-spanning channel complexes.

Visualisation of peptide–detergent complexes using native polyacrylamide gel electrophoresis (PAGE) revealed that DH$_6$PC (300 mM) induced the formation of higher-order oligomers (*Figure 1D*). A dominant species of ~45–50 kDa formed along with another less abundant assemblage in the region of ~60 kDa, although it is not possible to assign precise molecular weights by this method. However, this was consistent with the potential formation of hexameric and/or heptameric oligomers, confirming that dimers were not the only multimeric conformation adopted by free M protein.

## ZIKV M peptides exhibit acid-enhanced channel activity with sensitivity to rimantadine

Given the formation of higher-order oligomers, we next assessed whether M peptides exhibited membrane permeabilisation using an indirect assay for channel formation, based upon the release and unquenching of fluorescent dye from liposomes. This assay has been used to characterise the activity of several other viroporins and can be adapted to both explore aspects of channel biochemistry as well as identifying small molecule inhibitors of channel activity (*Foster et al., 2011*; *StGelais et al., 2009*; *Wetherill et al., 2012*; *Carter et al., 2010*; *Foster et al., 2014*; *Griffin et al., 2008*; *Scott et al., 2020*; *Shaw et al., 2020*; *StGelais et al., 2007*; *Walter et al., 2016*).

Dose-dependent dye release occurred using sub-micromolar concentrations of M peptide, consistent with channel forming activity (*Figure 2A*); 780 nM was chosen as an optimal concentration for ensuing investigations. Interestingly, mildly acidic pH conditions increased M-mediated release of carboxyfluorescein (CF), reminiscent of IAV M2 (*Scott et al., 2020*), hepatitis C virus (HCV) p7 (*StGelais et al., 2009*; *Atkins et al., 2014*), and human papillomavirus type 16 (HPV16) E5 (*Wetherill et al.,*

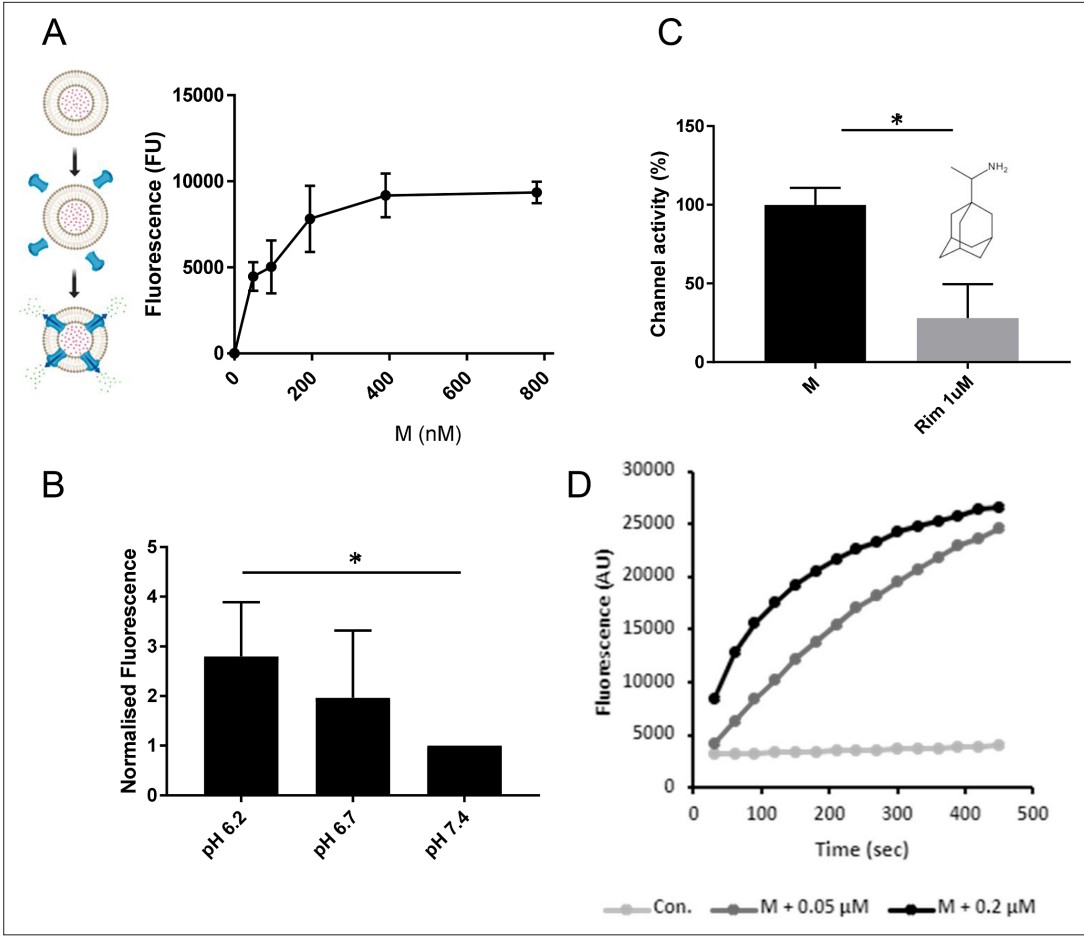

**Figure 2.** M induces rimantadine-sensitive membrane permeability, augmented by reduced pH. (**A**) Titration of Dimethyl Sulphoxide (DMSO)-reconstituted M peptide in endpoint liposome carboxyfluorescein (CF) release assay. Graph represents a single biological repeat representative of at least three others, comprising triplicate technical repeats at each concentration. Error bars show standard deviation. (**B**) Modified endpoint CF assay undertaken with altered external buffer pH as indicated. CF content of re-buffered clarified assay supernatants were detected as a single endpoint measurement. Data comprise three biological repeats for each condition, error bars show normalised standard error, *$p \leq 0.05$, Student's *t*-test. (**C**) Effect of rimantadine (1μM) upon M activity. Results are from three biological repeats normalised to 100% activity for solvent control. Error bars show adjusted % error. *$p \leq 0.05$, Student's *t*-test. (**D**) Real-time CF assay data for M peptides in 'virion-sized' liposomes, produced by extrusion through a 0.05-μm, rather than a 0.2-μm filter.

The online version of this article includes the following source data for figure 2:

**Source data 1.** Prism or Excel files containing summary endpoint (**A–C**) or real-time (**D**) fluorescence data for dye release assays.

*2012*) in similar experiments (***Figure 2B***). Such conditions mimic those found within endocytic vesicles, consistent with M channels potentially being activated during virus entry.

We tested whether M channel activity was sensitive to inhibitory small molecules, indicative of the formation of structurally ordered complexes. Prototypic channel blockers can be used to identify druggable sites within viroporin complexes as a result of their binding promiscuity. Regions so identified are then amenable to the development of improved small molecule inhibitors (***Foster et al., 2014***; ***Scott et al., 2020***; ***Shaw et al., 2020***). One such prototype, rimantadine (1 μM), caused a significant reduction in M channel activity (***Figure 2C***), consistent with the presence of a viable binding site within a soluble, folded M channel complex.

Finally, we tested whether M peptides were able to form channels within membranes exhibiting significant curvature, such as that seen within an enveloped virion. Liposomes extruded using a 0.05-μM filter provided a similar level of M-induced dye release, albeit with a slightly reduced kinetic, likely due to the increased molar ratio of resulting liposomes to peptide (*Figure 2D*).

## Rimantadine inhibits ZIKV infectivity in cell culture

We hypothesised that if M protein-mediated channel activity observed in vitro was relevant during the ZIKV life cycle, then rimantadine would exert an antiviral effect. We therefore tested the ability of rimantadine to inhibit ZIKV infection of Vero cells, adding the drug during infection and for the ensuing 24 hr period prior to analysis. Rimantadine exerted a dose-dependent reduction in ZIKV infection, assessed by both the number of cells infected and ensuing viral protein production (*Figure 3A–C*). Infected cell numbers underwent statistically significant reductions at rimantadine concentrations of 10 μM and above. Importantly, rimantadine affected neither cell viability, nor the uptake of fluorescently labelled (Epidermal Growth Factor) EGF as a proxy marker for clathrin-mediated endocytosis over the concentration range tested (*Figure 3—figure supplement 1*).

We next added rimantadine (80 μM) during different stages of the infection process, measuring drug effects by a (Baby Hamster Kidney) BHK cell plaque-reduction assay, to determine the approximate stage of the ZIKV life cycle affected. Statistically significant reductions of infectivity occurred when rimantadine was present during infection, with more pronounced effects upon pre-incubating cells with the drug (*Figure 3D*). Measurable, but non-significant reductions also occurred upon adding rimantadine only prior to infection, or during and following infection. Taken together, rimantadine affected an early stage of the virus life cycle, coincident with virus entry.

## MD supports that M protomers adopt a double *trans*-membrane topology

The structure of M within mature ZIKV and other *Flavivirus* virion cryo-EM reconstructions is a dimeric complex of double-TMD protomers, closely associated with the TMDs of adjacent E dimers (*Sirohi et al., 2016*). The relatively short TMDs within M cause a pinching of the virion membrane bilayer, reducing its thickness by several nm (*Sevvana et al., 2018*; *Kuhn et al., 2002*; *Mukhopadhyay et al., 2003*).

M is highly conserved between ZIKV isolates (*Figure 4A*), and regions including the linker region between the first and second alpha helices are shared with other *Flaviviruses* (*Figure 1—figure supplement 2*). Where sequence conservation is absent, secondary structure and amino acid similarity are maintained. However, four different TMD prediction software packages revealed disparate outputs for the region known to form the second and third helices within cryo-EM reconstructions, with two favouring the formation of an extended single TMD (*Figure 1—figure supplement 2*). Interestingly, the MEMSAT-SVM package not only predicted a double TMD topology, but also that helix 3 displayed properties consistent with being pore lining (*Figure 1—figure supplement 2*).

Thus, we undertook MD simulations to address whether M in the absence of potentially stabilising interactions with E dimers might prefer a double, or a single TMD conformation. Monomeric M protein was placed within model POPC bilayers (*Figure 4C, D*, *Figure 4—figure supplements 1 and 2*), starting as either the cryo-EM-derived conformer, or one where the two TMDs were straightened into a membrane-spanning helix containing a small unstructured region between helices 2 and 3 (*Figure 4C, D*, *Figure 4—figure supplements 1–3*). Simulations of the two species within a POPC lipid bilayer revealed, unsurprisingly, that the double helix remained stable over time consistent with the parental cryo-EM structure (*Figure 4D*). In contrast, the single helix form rapidly began to fold back towards the membrane, with helix 3 partially overlaying the phosphate groups of one monolayer. Hence, this form of the peptide was clearly unstable compared to the cryo-EM structure and we considered it likely that potential higher-order M oligomers also comprised double-TMD protomers.

## MD simulations favour the formation of compact hexameric channel complexes

Observations from native PAGE and electron microscopy led us to construct hexameric, and heptameric models of putative M channels with helix 2 or 3 lining the channel lumen (*Figure 5*, *Figure 5—figure supplements 1 and 2*, Appendix); helix 3 was predicted to display pore-lining properties based upon

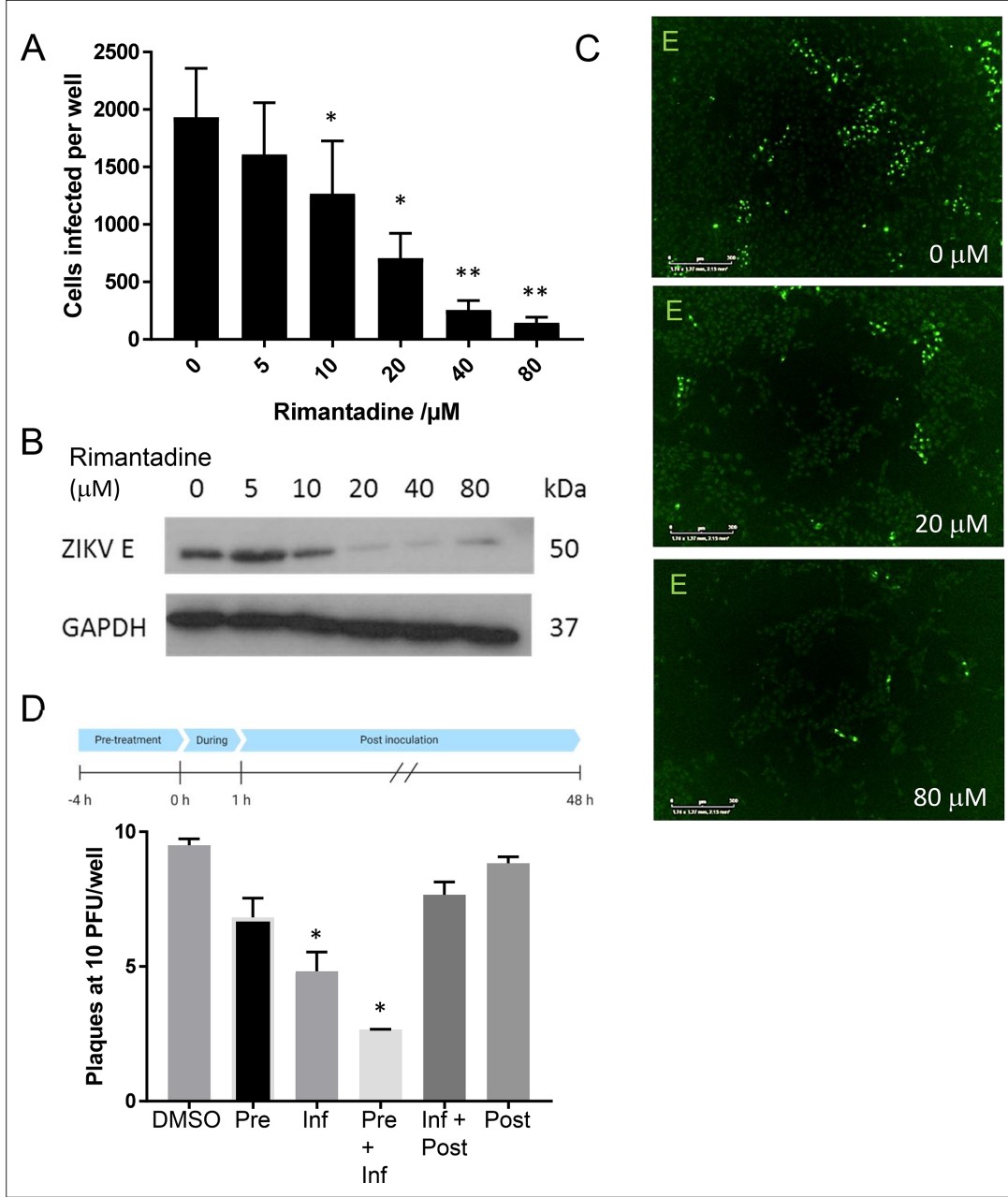

**Figure 3.** Rimantadine blockade of ZIKV infection. (**A**) Vero cells were infected at a multiplicity of infection (MOI) of 0.1 pfu/cell with increasing rimantadine concentrations in 96-well plates. Infectious units were quantified 48hr post-infection using an IncuCyte Zoom to count ZIKV infected cells stained with antibody to ZIKV E and an Alexafluor 488nm secondary antibody. Counts were averaged over four image panels per well and results are from three biological replicates performed in triplicate. Error bars represent standard error of the mean between experiments. *$p \leq 0.01$, **$p \leq 0.001$, paired Student's $t$-test relative to solvent control. (**B**) Representative anti-E western blot of $n = 3$ biological repeats of 6-well plate experiments run in parallel to fluorescence-based assays. Infections and timings as for A. (**C**) Example representative fluorescence images from IncuCyte analysis shown in A, scale bars represent 500μm. (**D**) Plaque-reduction assay using BHK21 cells infected with 10 pfu/well in the presence of 80μM rimantadine during different stages of infection shown by schematic. $N = 2$ biological replicates. Error bars represent standard error of the mean between experiments, *$p \leq 0.01$, paired Student's $t$-test relative to solvent control.

The online version of this article includes the following source data and figure supplement(s) for figure 3:

**Source data 1.** Prism files summarising fluorescence focus assays for rimantadine-mediated inhibition during

*Figure 3 continued on next page*

*Figure 3 continued*

multi-step infection, or time of addition assays.

**Figure supplement 1.** Controls for off-target effects of rimantadine during infectious ZIKV culture.

**Figure supplement 1—source data 1.** Prism files with data for rimantadine MTT assays and fluorescent EGF uptake assays.

MEMSAT-SVM predictions (*Figure 1—figure supplement 2*), so this class of models was the principal focus. Two different hexameric arrangements were modelled (*Figure 5*, *Figure 5—figure supplements 1–3*): in the first arrangement, helix 1 was orientated almost perpendicular to the channel pore with helix 3 facing towards the pore (termed 'radial', *Figure 5E*, *Figure 5—figure supplements 1–3*). For the second model, the protein was rotated clockwise by 20 degrees to significantly increase the number of inter-protomer contacts within a more compact structure (termed 'compact', *Figure 5A–E*, *Figure 5—figure supplements 1–3*). The second conformation included interactions between the extra-membranous helix 1. Anti-clockwise rotation was not possible due to stearic hindrance. Accordingly, removal of helix 1 caused the collapse of stable pores within 50 ns in each of three simulations (*Figure 5—figure supplement 1*), demonstrating its likely importance in the formation of a stable channel complex.

We also generated radial and compact hexamer and heptamer models, including with helix 2 lining the pore, ensuring that other potential conformations were explored (see Appendix). However, hexamer models where helix 2 lined the pore collapsed rapidly upon simulation, whereas heptamers lined by either helix showed an inability to close in the majority of simulations (Appendix). Moreover, radial hexamers lined by helix 3 either closed rapidly within the simulation (6, 40 ns) or not at all (>200 ns). In the two channels that closed, one of the subunits moves to the centre of the channel to occupy the pore (Appendix). This again was inconsistent with the formation of a stable channel complex. Notably, no restraints were placed upon secondary structures during atomistic simulations of channel complexes, providing added confidence of their relative stability.

Multiple atomistic simulations for each condition supported that the compact hexameric model with helix 3 lining the lumen comprised the most stable and viable representation of a putative channel structure formed by M protein (*Figure 5*). The compact channel structure retained a lumen with a radius of 5.23 Å at its narrowest point, lined by Thr57, Val61, Ile62, Val65, Met66, Leu68, and Leu69 (*Figure 5B*). Upon 200 ns atomistic simulation within a model POPC bilayer, channels initially allowed the formation of a water column, but then closed at Leu64 and Leu68, despite the former not facing the lumen at the beginning of the simulation (*Figure 5C–E*). Channel closing first occurred at 70, 80, and 176 ns within three separate simulations, with channels remaining closed for the remainder of the simulation (*Figure 5C, D*). Thus, despite the starting position of this model possessing a relatively wide lumen compared with the diameter of a water column (~1.15 Å), compact M hexamer models lined by helix 3 displayed spontaneous closure, consistent with a reasonable physiological representation of a membrane channel. Interestingly, despite the effect of acidic external pH upon M channel activity in vitro (*Figure 2B*), mimicking such an environment by protonation of a potential candidate pH sensor, His28 within helix 1, had little appreciable effect upon channel opening during simulations. This was confirmed by His28Ala mutant peptides responding to acid pH identically to wild-type in vitro (*Figure 5—figure supplement 4*). Thus, other ionisable residues within the M protein likely mediate acid-enhanced activity in vitro.

## Identification of two potential druggable sites within hexameric M complexes

We hypothesised that MD-tested compact hexamer models might be sufficiently accurate to both characterise and improve upon the inhibitory action of rimantadine. Whilst this drug exerted a genuine antiviral effect, its relatively low potency in cell culture was reminiscent of its promiscuous activity against other viroporins (*Wetherill et al., 2012*; *Griffin et al., 2008*; *Scott et al., 2020*). Nevertheless, the action of rimantadine both in vitro and in vivo implied that at least one physiologically relevant druggable binding site exists within the M channel complex.

Docking of rimantadine into the compact M hexamer model revealed an energetically favourable interaction with a lumenal binding site, close in proximity to the region in which channel closure

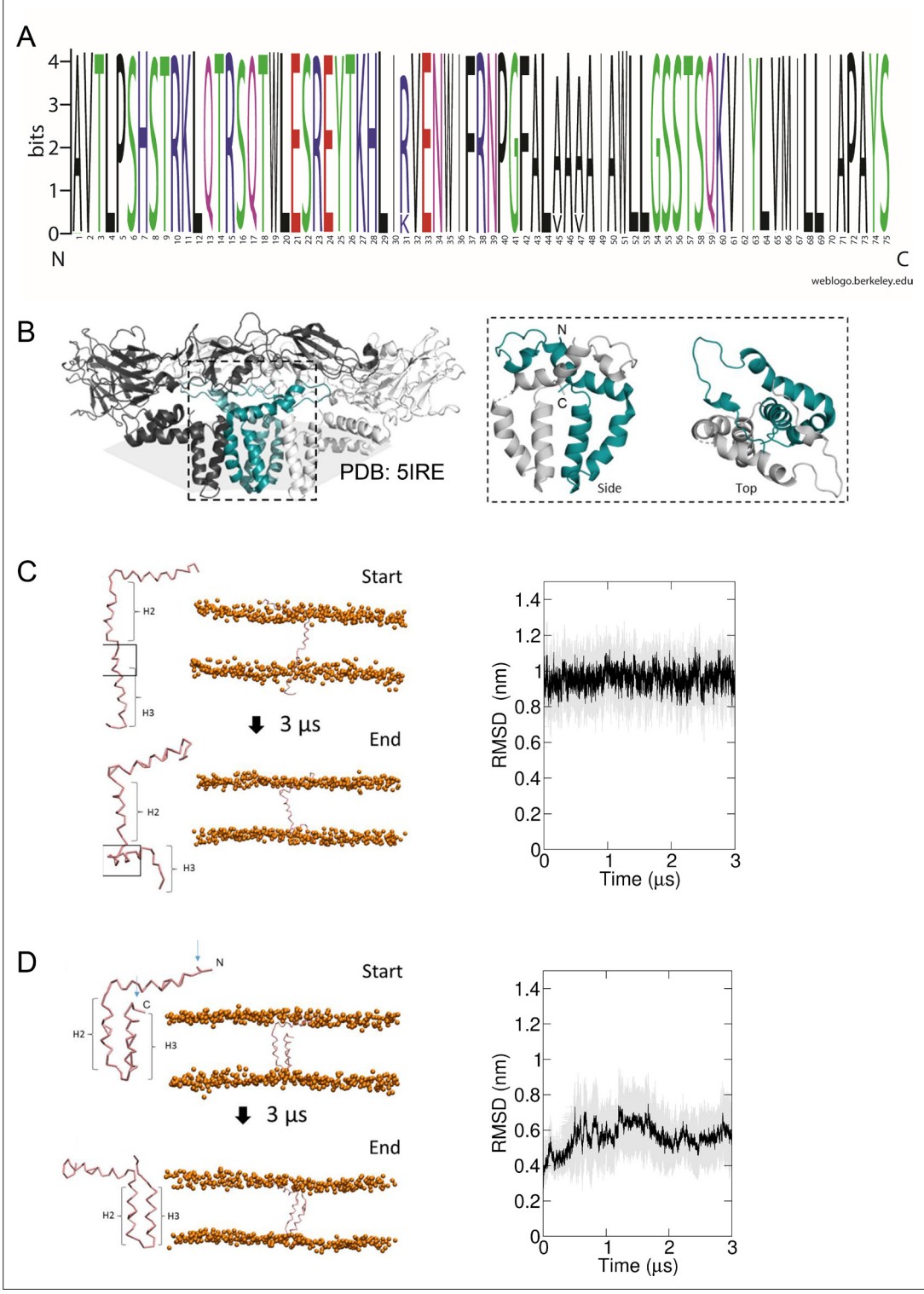

**Figure 4.** Predicted membrane topology for M protomers. (**A**) M sequence 'logo' (https://weblogo.berkeley.edu/logo.cgi) showing relative conservation at each amino acid position. 929 sequences were retrieved by a Uniprot search for 'Zika virus', then aligned using CLUSTAL Omega. Alignments were exported to Jalview where removal of partial/irrelevant sequences resulted in analysis of~700 sequences. Conservation is high (>95%) across the majority of the protein, with the exception of Arg31, Ala45, and Ala47, but even these remain present within>80% sequences. (**B**) Image of an energy minimised M protomer taken from PDB: 5IRE within a simulated hydrated lipid membrane system (see methods). (**C**) Straightened, single TM M protomer inserted into a POPC bilayer and

*Figure 4 continued on next page*

*Figure 4 continued*

subjected to coarse-grained (CG) simulations for 3 μs, running from start (top) to finished (bottom) pose. (root mean square deviation) RMSD over time (five repeat simulations, standard deviations in grey) showed a high degree of movement and flexibility exhibited by this conformation (~1nm throughout), as the protein appears to attempt moving the C-terminal helix (H3) back along the plain of the membrane. Into close proximity of the bilayer. (**D**) By comparison, the cryo-EM-derived conformer structure was simulated in a model bilayer as for C. This conformation remained stable for the duration of the simulation (five repeat simulations, standard deviations in grey).

The online version of this article includes the following source data and figure supplement(s) for figure 4:

**Source data 1.** Folder containing all ZIKV sequences obtained from Uniprot, aligned in ClustalW and visualised using Jalview.

**Figure supplement 1.** Simulation data for straightened M protein monomer in POPC membranes.

**Figure supplement 2.** Simulation data for double trans-membrane cryo-EM derived M conformer in POPC membranes.

**Figure supplement 3.** RMSD over time from simulations shown in ***Figure 4C, D***, highlighting each of the five repeats as different colours.

---

occurred during simulations, termed 'L1' (***Figure 6A***, ***Figure 6—figure supplement 1***). The adamantyl cage of rimantadine made predicted hydrophobic contacts with Val61, Ile62, Leu64, Val65, and Leu68, leaving the methyl group and amine solvent exposed, projecting into the channel lumen (***Figure 6— figure supplement 1***). The same site also underwent predicted interactions with other adamantyl-containing molecules, including amantadine, which adopted a similar predicted binding mode to rimantadine (***Figure 6—figure supplement 1***). However, both methyl- and acetyl-rimantadine reversed the orientation of the adamantyl cage such that the polar group interacted with lumenal residues (***Figure 6—figure supplement 1***).

Reassuringly, the lumenal rimantadine site (L1) was the most favourable in terms of druggability score for the compact hexamer model, assessed using the SiteMap programme (Maestro, Schrödinger). Two other potential lumenal sites were also identified with lower predicted druggability scores (L2, L3, ***Figure 6A***). However, inspection of the M channel model identified an additional region upon the membrane-exposed periphery that could also undergo interactions with small molecules (P1, ***Figure 6A***). Such binding sites are less likely to be identified computationally due to the high solvent exposure of potential ligands, yet such sites have been successfully targeted for other viroporins, including HCV p7 (***Scott et al., 2020***; ***Shaw et al., 2020***).

To explore the potential relevance of the P1 site, we exploited a defined series of compounds developed targeting a similar site upon the HCV p7 channel complex. The planar structure of these compounds is more compatible with the shape of the P1 site compared to L1, as supported by the predicted in silico modelling. Encouragingly, two of the seven compounds tested displayed inhibitory effects versus M in dye release assays, with JK3/42 being most active (***Figure 6—figure supplement 1***). Interestingly, adamantane derivatives with additional R groups were ineffective at targeting M activity (***Figure 6—figure supplement 1***), potentially due to the limited size of the L1 cavity. Thus, we investigated the possibility that more than one distinct druggable binding site existed within M complexes.

## In silico screening enriches for improved repurposed M channel inhibitors

We investigated whether in silico screening could identify and enrich for compounds predicted to interact with L1 and/or P1, with a view to developing two distinct yet complementary inhibitor series. L1 was defined as the site comprising Val61, Leu64, Val65, and Leu68, and P1 by Tyr63, Leu64, Val65, Met66, Ile67, and Leu68. The two sites effectively represented the internal and external face of a single region within the channel complex, which is also involved during channel closing in simulations and centres upon Leu68. The convergence of these three regions was suggestive of their importance in being able to influence channel gating (***Figures 5D, E and 6A***).

To determine whether compounds with good drug-like properties could target the L1 and P1 sites, we employed a screening library comprised of FDA-approved, generic, and other compounds with

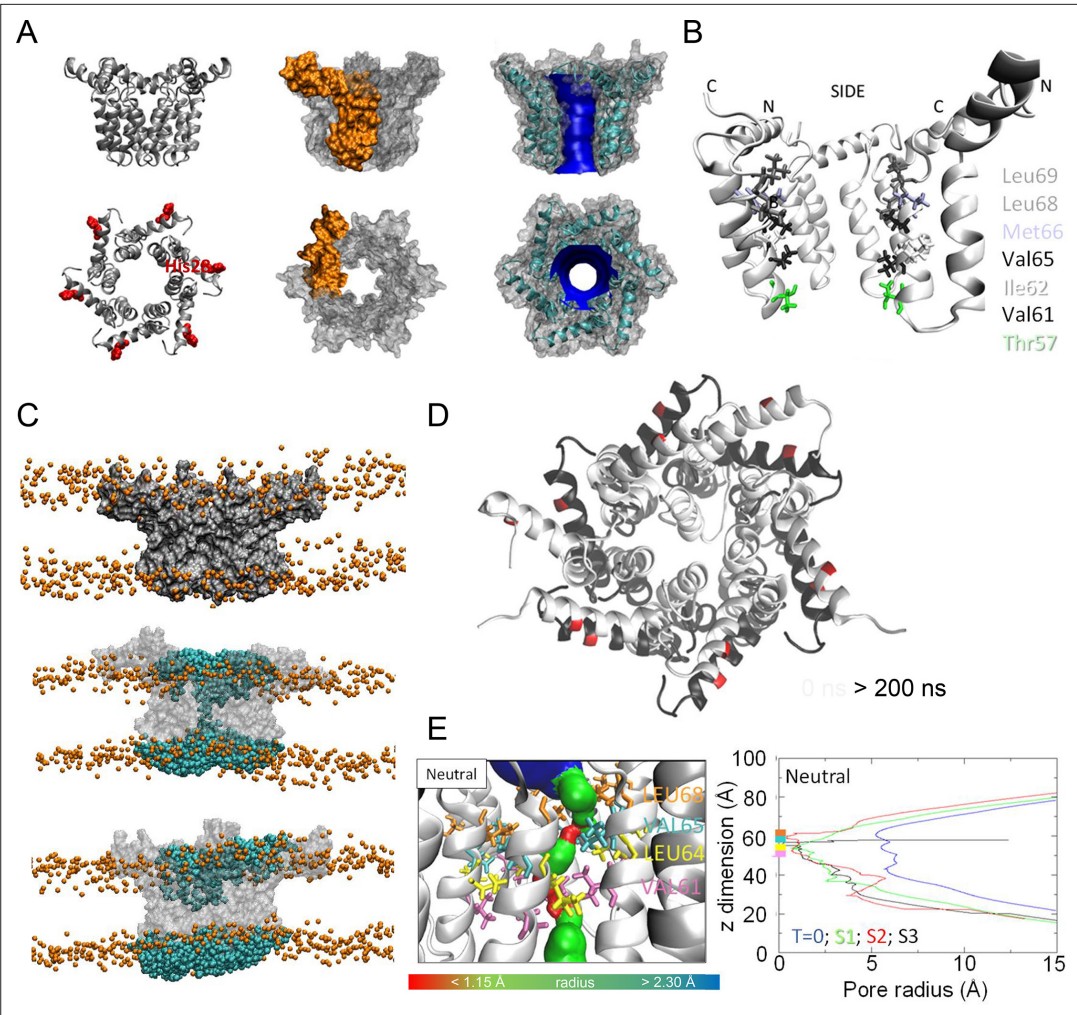

**Figure 5.** Simulated hexameric M channel complex in a compact formation, lined by helix 3. (**A**) Assembly of protomers into compact hexamer model using Maestro. Left – ribbon diagram illustrating position of His28 (red) in helix 1 for orientation; middle – space filling model showing individual protomer (gold); right – space filling model showing pore diameter using HOLE. (**B**) Cutaway of channel viewed from the side showing pore-lining residues predicted for the compact hexamer from the side. (**C**) Snapshots of compact hexamers in POPC bilayers from a representative atomistic simulation showing the energy minimised system at *t* = 0 (top), formation of a conductive water column during early times and eventual closing of the channels. (**D**) Compact hexameric complex over a 200-ns atomistic simulation overlaying start (white) and endpoint (black) conformations. (**E**) HOLE profiles for lumenal aperture showing differences between starting configuration (blue line) and final structures over three separate 200 ns atomistic simulations (green, red, and black) at neutral external pH.

The online version of this article includes the following source data and figure supplement(s) for figure 5:

**Source data 1.** Folder containing MP4 files of two 'Compact hexamer, H3' simulations.

**Figure supplement 1.** MD simulations of hexameric conformers under different conditions.

**Figure supplement 2.** MD simulation data for "radial" hexameric conformers.

**Figure supplement 3.** MD simulation data for "compact" hexameric conformers.

**Figure supplement 4.** Exploration of mimicking acidic external pH via dual His28 protonation (His28++).

**Figure supplement 4—source data 1.** Prism file for dye release assay data.

proven biological activity. In silico screening of 1280 compounds (TocrisScreen, https://www.tocris.com/product-type/tocriscreen-compound-libraries) was undertaken using Glide (Schrödinger) and a rank-order list generated for each site following attrition and removal of compounds common to both sites. A short list of 50 compounds targeting each site was generated for subsequent validation

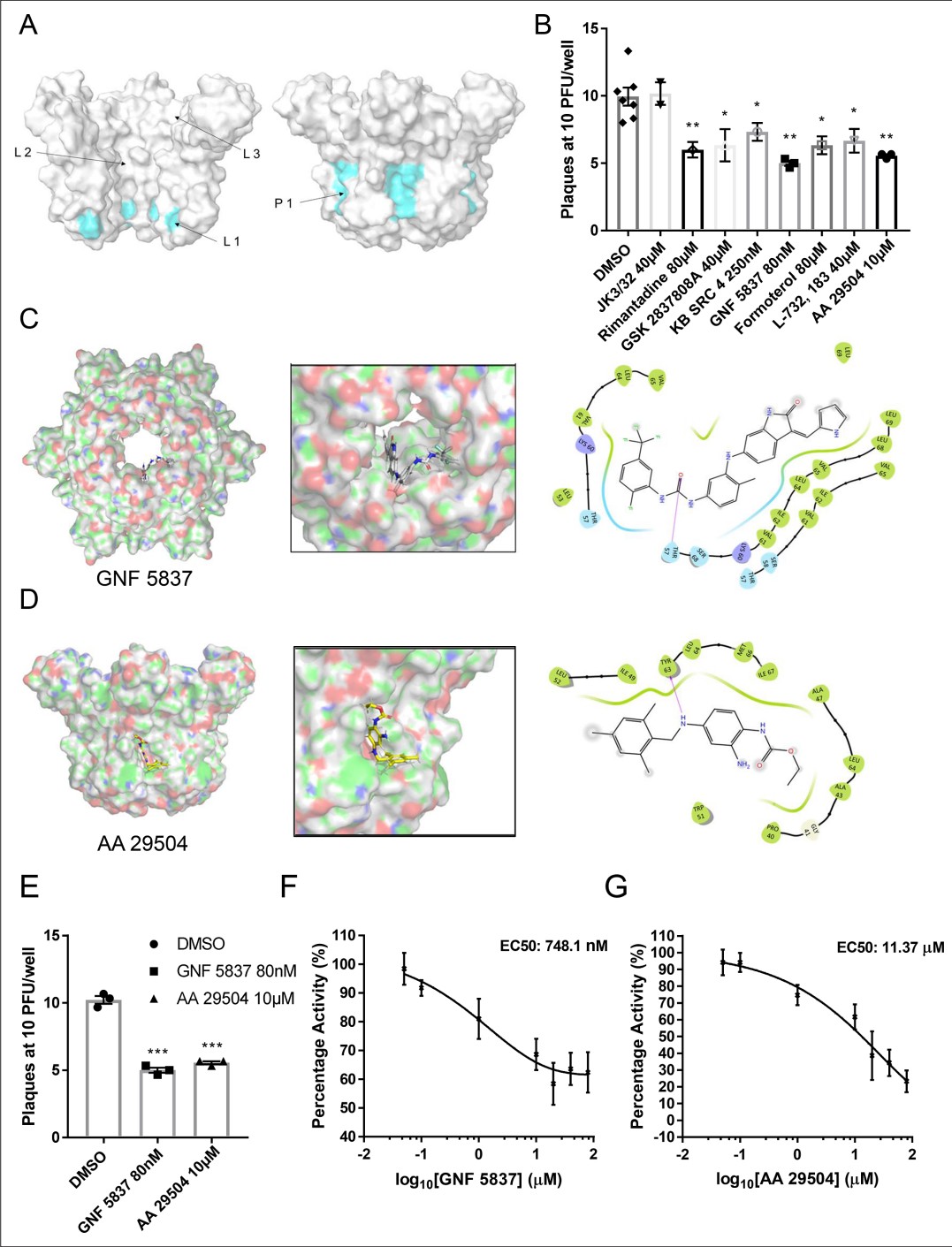

**Figure 6.** Inhibition of ZIKV replication by in silico enriched repurposed drugs. (**A**) Compact M channel complex models were assessed using the SiteMap package within Maestro for cavities corresponding to high druggability scores. Three sites were found within the lumen (L1–L3),with L1 retaining the most favourable score. An additional site upon the membrane-exposed channel periphery was also evident (P1),yet would not score highly in SiteMap due to solvent exposure of compounds during simulated docking. Notably, both L1 and P1 reside within close proximity to the predicted gating region near Leu64. (**B**) Compounds selected to bind the L1 and P1 sites in silico and screened in vitro (***Figure 6—figure supplement 2***) were tested for effects in ZIKV culture, using a BHK21 cell plaque-reduction assay. Compounds were used at concentrations shown not to evoke cellular toxicity by MTT assays (***Figure 6—figure supplement 3***). Experiments are biological triplicates with error bars representing the standard error of the mean. **p ≤ 0.01, *p ≤ 0.05, Student's *t*-test. (**C**) Predicted binding pose of GNF 5837 at L1 site generated using Glide (Maestro, Schrodinger) with associated binding site interactions. (**D**) As C, but for AA

*Figure 6 continued on next page*

*Figure 6 continued*

29504. (**E**) Example biological repeat with triplicate technical repeats for the two top compounds targeting the L1 and P1 sites, GNF 5837 (L1)and AA 29504 (P1). (**F**) In vitro dye release assays showing 8-point log$_2$-fold titration of GNF 5837 enabling calculation of 50% effective concentration (EC$_{50}$). (**G**) As F, for AA 29504.

The online version of this article includes the following source data and figure supplement(s) for figure 6:

**Source data 1.** Prism and excel files for both plaque assay and in vitro dye release EC$_{50}$ experiments using lead lumenal and peripherally targeted M inhibitors.

**Figure supplement 1.** Lack of inhibitory activity for other adamantane derivatives versus M in carboxyfluorescein (CF) release assays.

**Figure supplement 1—source data 1.** Prism file for liposome dye release assay using M peptides with adamantane analogues and JK series compounds.

**Figure supplement 2.** Screening data for lumenally (**A**) and peripherally targeted (**B**) M-binding compounds, enriched using in silico high-throughput screening (HTS). The top 50 compounds based upon predicted binding scores and drug-like properties derived from the TOCRIS repurposing library were tested versus M at 1μM using carboxyfluorescein (CF) release assays. Results show the average of three biological repeats, with error bars representing the standard error of the mean.

**Figure supplement 2—source data 1.** Prism file with data for repurposing library screen using dye release assays.

**Figure supplement 3.** Effects of screening hits upon cell metabolic activity measured by MTT assay in Vero cells.

**Figure supplement 3—source data 1.** Prism file with data for hit compound MTT assays in BHK cells.

**Figure supplement 4.** Predicted binding poses and residue interactions for peripherally targeted hit compounds.

**Figure supplement 5.** Predicted binding poses and residue interactions for lumenally targeted hit compounds.

in vitro using dye release assays (*Tables 1 and 2*). Note, predicted docking scores are not directly comparable between P1 and L1 owing to binding penalties incurred at the P1 site discussed above.

Dye release assay screens (96-well format) were conducted using compounds at a concentration of one micromolar, with positive hits defined as compounds exerting a 50 ± 5% decrease in M activity (*Figure 6—figure supplement 2*). Twenty-four hits were identified from the L1-targeted short list, and 15 for the P1 site.

We selected three compounds from each list, based upon both potency and commercial availability, for validation in plaque-reduction assays. As the components of the repurposing library predominantly target cellular factors, MTT assays were used to determine non-toxic maximal concentrations for each compound (*Figure 6—figure supplement 3*). Testing at these concentrations revealed that each of the compounds exerted a statistically significant reduction upon ZIKV infectivity, with the most effective showing sub-micromolar potency (*Figure 6B*), whereas the inactive JK3/32 compound showed no detectable antiviral effects.

Two L1-selected compounds, KB SRC 4 and GNF 5837 displayed antiviral efficacy at 250 and 80 nM, respectively, whereas the most effective P1-targeted compound, AA 29504, was active at 10 μM. Examination of the predicted binding pose for these compounds at L1/P1 revealed that both molecules underwent additional interactions with regions adjacent to the core-binding sites (*Figure 6C, D*, *Figure 6—figure supplements 4 and 5*). GNF 5837, an inhibitor of the tropomyosin receptor kinase (TRK) family, was predicted to make extended contacts within the lumen, including with two of the predicted rimantadine-binding sites, stabilised by H-bonding between a carbonyl and Thr57 (*Figure 6C*). These leading hit compounds were re-tested in a second round of plaque-reduction assays, confirming efficacy (*Figure 6E*) and displayed in vitro EC$_{50}$ values at similar orders of magnitude as their observed potency in cell culture, namely sub-micromolar for GNF 5837, and approximately 10 μM for AA 29504 (*Figure 6F, G*).

The considerable enrichment for potency, compared to rimantadine, seen amongst compounds selected by in silico screening supports the validity and utility of structure-guided models in their potential use as templates for rational drug development.

## M channel inhibitors reduce the specific infectivity of ZIKV virions

It was important to establish whether M-targeted compounds might exert off-target effects upon virion stability, or other aspects of virus entry. Frustratingly, the role of (pr)M during virion egress precludes the efficient generation of *Flavivirus* envelope pseudotype systems in their absence, an

**Table 1.** Top 50 lumenally targeted compounds from the TOCRIS screen library, ranked by glide score.

| 'Rank' | Compound name | Tocris ID | Glide gscore |
|---|---|---|---|
| 1 | FR 139317 | 1210 | −10.227 |
| 2 | ZCL 278 | 4794 | −9.294 |
| 3 | Elinogrel | 5316 | −8.829 |
| 4 | Taxol | 1097 | −8.711 |
| 5 | UK 356618 | 4187 | −8.401 |
| 7 | TC-1 15 | 4527 | −8.216 |
| 8 | AS 2034178 | 5035 | −8.032 |
| 9 | CP 775146 | 4190 | −7.963 |
| 10 | GW 6471 | 4618 | −9.092 |
| 11 | Pravastatin sodium salt | 2318 | −7.825 |
| 12 | Fluvastatin sodium | 3309 | −7.717 |
| 13 | TC NTR1 17 | 5087 | −7.683 |
| 14 | VER 155008 | 3803 | −7.686 |
| 15 | AMG PERK 44 | 5517 | −7.618 |
| 16 | Glibenclamide | 0911 | −7.537 |
| 17 | Argatroban | 1637 | −7.454 |
| 18 | KB SRC 4 | 4660 | −7.444 |
| 19 | GBR 12909 dihydrochloride | 0421 | −7.965 |
| 20 | (±)-NBI 74330 | 4528 | −7.366 |
| 21 | CU CPT 4a | 4883 | −7.331 |
| 22 | A 887826 | 4249 | −7.315 |
| 23 | SR 2640 hydrochloride | 1804 | −7.296 |
| 24 | NSC 74859 | 4655 | −7.258 |
| 25 | RWJ 67657 | 2999 | −7.217 |
| 26 | Lu AA 47070 | 4783 | −7.659 |
| 27 | Edaglitazone | 4784 | −7.176 |
| 29 | GSK 1562590 hydrochloride | 5110 | −7.11 |
| 30 | Flurizan | 4495 | −7.098 |
| 31 | GW 9508 | 2649 | −7.096 |
| 32 | GSK 269962 | 4009 | −7.051 |
| 33 | AC 5216 | 5281 | −6.986 |
| 34 | DBZ | 4489 | −6.973 |
| 35 | PF 04418948 | 4818 | −6.957 |
| 36 | GSK 2837808A | 5189 | −6.942 |
| 37 | Sal 003 | 3657 | −6.935 |
| 38 | PD 173212 | 3552 | −7.029 |
| 39 | NTRC 824 | 5438 | −9.012 |
| 40 | ONO AE3 208 | 3565 | −6.889 |
| 41 | RS 17053 hydrochloride | 0985 | −6.88 |
| 42 | Pitavastatin calcium | 4942 | −6.864 |

*Table 1 continued on next page*

*Table 1 continued*

| 'Rank' | Compound name | Tocris ID | Glide gscore |
|---|---|---|---|
| 43 | L-161,982 | 2514 | −6.858 |
| 44 | AMN 082 dihydrochloride | 2385 | −6.871 |
| 45 | TC-N 1752 | 4435 | −6.815 |
| 46 | PF 431396 | 4278 | −6.799 |
| 47 | GNF 5837 | 4559 | −6.754 |
| 48 | KS 176 | 4169 | −6.731 |
| 49 | Sarpogrelate hydrochloride | 3739 | −6.769 |
| 50 | GKA 50 | 5133 | −6.689 |
| N1 | Mifepristone | 1479 | 0.286 |

otherwise useful tool in this regard. As an alternative, compound-treated ZIKV particles were compared to solvent controls following ultracentrifugation and separation using a continuous iodixinol gradient. This not only allows characterisation of specific infectious species, but also sequesters small molecules at the top, lower-density range of the gradient, preventing incidental exposure of target cells when inoculating with virus-containing fractions, and so minimising potential cell off-target effects.

Interestingly, ZIKV infectivity profiles contained two peaks of infectivity when samples were diluted 1:10 onto target cells: a low-density peak at ~1.11 g/ml, and a high-density peak at 1.13–1.14 g/ml. However, detection of infectivity within the low-density peak diminished upon dilution at 1:100 or 1:1000, and this was not the case for the high-density peak (*Figure 7—figure supplement 1*). Interestingly, the distribution of the ZIKV E protein was considerably more abundant within lower-density fractions compared to the main infectivity peak at higher density (*Figure 7A*). Drug treatment specifically reduced infectivity (measured at 1:10 supernatant dilution to maintain both species) within the high-density infectious fraction compared to controls; the lower-density fractions were unaffected (*Figure 7A, B*). Importantly, as well as a global loss of infectivity, specific infectivity within the high-density fraction normalised by (reverse transcriptase quantitative polymerase chain reaction) RT-qPCR for ZIKV RNA (E gene target) was also significantly reduced (*Figure 7C*), meaning that intact virions were rendered less infectious following exposure to inhibitors rather than being physically disrupted via non-specific 'virolysis'. Moreover, based upon previous studies, the concentration of inhibitors added to target cells from high-density fractions would have been negligible, excluding artefactual effects upon cell entry.

## Rimantadine prevents ZIKV viraemia in vivo

We assessed rimantadine antiviral effects in vivo using a preclinical ZIKV infection model (*Pingen et al., 2016*) comprising C57BL/6 mice and transient blockade of the interferon alpha receptor type 1 (IFNAR1). This allows the establishment of robust infection within an immunocompetent host and does not lead to the severe sequelae seen within immunodeficient systems. Moreover, the model incorporates improved physiological relevance via concomitant biting at the inoculation site by female *A. aegypti* mosquitos, which enhances the efficiency of infection (*Figure 8A*).

Ten C57BL/6 mice received a sub-cutaneous dose of 20 mg/kg rimantadine (or carrier control), 30 min prior to ZIKV infection ($10^3$ plaque forming units, pfu) by injection into the dorsal side of one hind foot, immediately following exposure to up to five mosquito bites. A second bolus of rimantadine was administered 5 hr post-infection. Animals were sacrificed and processed 24 hr post-infection, harvesting tissue from the inoculation site as well as blood via cardiac puncture. RT-qPCR analysis confirmed equivalent copies of viral RNA present within tissues at the inoculation site (*Figure 8B*), whereas viraemia was dramatically reduced within rimantadine-treated animals, measured by BHK cell plaque assay (*Figure 8C*). Thus, rimantadine exerted an antiviral effect in vivo that prevented dissemination of the virus, consistent with cell culture observations, and supporting the druggability of M in vivo.

**Table 2.** Top 50 peripherally targeted compounds from the TOCRIS screen library, ranked by glide score.

| 'Rank' | Compound | Cat No | Glide gscore |
|---|---|---|---|
| 1 | RWJ 21757 | 2719 | −6.475 |
| 2 | Ferrostatin 1 | 5180 | −6.309 |
| 3 | AA 29504 | 3972 | −6.31 |
| 4 | L-732,138 | 0868 | −6.167 |
| 5 | API-2 | 2151 | −6.158 |
| 6 | 5-BDBD | 3579 | −6.13 |
| 7 | LY 225910 | 1018 | −6.082 |
| 8 | Formoterol hemifumarate | 1448 | −6.065 |
| 9 | TC-S 7006 | 5240 | −5.979 |
| 10 | TCS 2210 | 3877 | −6.005 |
| 11 | Sumatriptan succinate | 3586 | −5.952 |
| 12 | MRS 3777 hemioxalate | 2403 | −6.964 |
| 13 | Thiamet G | 4390 | −6.079 |
| 14 | Abacavir hemisulfate | 4148 | −5.894 |
| 15 | DDR1-IN-1 | 5077 | −5.99 |
| 16 | Fexofenadine hydrochloride | 2429 | −5.859 |
| 17 | 6-Chloromelatonin | 0443 | −5.844 |
| 18 | GSK 0660 | 3433 | −5.864 |
| 19 | PCA 4248 | 0571 | −5.783 |
| 20 | Axtinib | 4350 | −5.765 |
| 21 | DSR 6434 | 4809 | −5.971 |
| 22 | Necrostatin-1 | 2324 | −5.756 |
| 23 | Trifluorothymidine | 4460 | −5.838 |
| 24 | Cilndipine | 2629 | −5.729 |
| 25 | Efondipine hydrochloride monoethanolate | 3733 | −5.693 |
| 26 | ITE | 1803 | −5.69 |
| 27 | L-165,041 | 1856 | −5.689 |
| 28 | EB 47 | 4140 | −6.033 |
| 29 | GSK 2830371 | 5140 | −5.657 |
| 30 | AZD 1480 | 5617 | −5.738 |
| 32 | Amlodipine besylate | 2571 | −5.638 |
| 33 | Melatonin | 3550 | −5.628 |
| 34 | Fludarabine | 3495 | −5.626 |
| 35 | PF 06447475 | 5716 | −5.609 |
| 36 | SU 6668 | 3335 | −5.608 |
| 37 | AZD 5438 | 3968 | −5.836 |
| 38 | SU 11274 | 4101 | −6.145 |
| 39 | FPL 64176 | 1403 | −5.543 |
| 40 | Sunitinib malate | 3768 | −5.543 |
| 41 | YK 4-279 | 4067 | −5.535 |

*Table 2 continued on next page*

*Table 2 continued*

| 'Rank' | Compound | Cat No | Glide gscore |
|---|---|---|---|
| 42 | Ralfinamide mesylate | 4029 | −5.678 |
| 43 | ML 298 hydrochloride | 4895 | −5.504 |
| 44 | FH 1 | 5254 | −5.485 |
| 45 | PLX 647 dihydrochloride | 5102 | −6.23 |
| 46 | GPi 688 | 3967 | −5.438 |
| 47 | CP 94253 hydrochloride | 1317 | −5.441 |
| 48 | CGP 57380 | 2731 | −5.44 |
| 49 | BW 723C86 hydrochloride | 1059 | −5.433 |
| 50 | LY 364947 | 2718 | −5.84 |

## Discussion

This work supports that the ZIKV M protein mediates membrane permeability in vitro through the formation of oligomeric structures, and that small molecules targeting this activity also decrease the infectivity of ZIKV at an early stage of the life cycle in culture. This is reminiscent of the IAV M2 protein, which responds to endosomal pH to allow protonation of the virion core, expediting virion uncoating (*Wharton et al., 1994*); we have recently demonstrated that HCV p7 plays a similar role (*Shaw et al., 2020*). Moreover, like M2, the activity of M is sensitive to both prototypic small molecules and rationally targeted ligands, preventing the spread of infection both in cell culture and in vivo. Thus, M channels represent a potential new target for antiviral therapies that could both limit disease severity and engender prophylactic use to disrupt transmission.

M resides as E-associated dimers within the mature virion (*Sirohi et al., 2016*), yet for M channels to assemble, higher-order structures must form in the virion membrane to generate an intact aqueous pore. We hypothesise that the rearrangement and formation of trimeric E complexes during acid-induced fusion may provide an opportunity for M to also adopt alternative conformations. Density corresponding to M is absent from cryo-EM reconstructions of acidified mature *Flavivirus* virions, which are themselves inherently unstable and require the binding of E-specific FAb fragments to preserve their architecture (*Zhang et al., 2015*). It is possible that asymmetric formation of relatively few channel complexes leads to a loss of signal upon application of icosahedral symmetry, or due to the overwhelming inherent symmetry of the particle. It would be of interest to apply symmetry expansion or focused classification techniques to acidified virions to investigate this possibility.

DENV M protein was previously proposed to function as a viroporin based primarily upon its size and hydrophobicity, yet conclusive evidence for a defined role within *Flavivirus* life cycles has been lacking. C-terminal M-derived peptides from DENV type 1 were shown to form cation-selective channels in suspended bilayers (*Premkumar et al., 2005*), with sensitivity to amantadine (10 μM) and higher concentrations of hexamethylene amiloride (100 μM). However, a subsequent study using a series of pr-M protein expression constructs with additional N-terminal signal peptides and myc epitope tags failed to demonstrate pH activated channel activity in *Xenopus laevis* oocytes (*Wong et al., 2011*). Nevertheless, in addition to the difficulties interpreting negative data, it was unclear whether proteins derived from these constructs underwent authentic proteolytic processing to enable trafficking of mature M protein to the oocyte membrane. Older studies showed that DENV replication in human peripheral blood leukocytes and Rhesus macaque kidney epithelial cells (LLC-MK2) was suppressed by amantadine and rimantadine, with maximal effect when drugs were administered concomitant with infection (*Koff et al., 1980*), reminiscent of findings herein.

Consistent with the formation an aqueous pore across the membrane, ZIKV M peptides migrated as higher-order structures when reconstituted in the membrane-mimetic detergent, DH$_6$PC. These corresponded to potential hexamers or heptamers, albeit estimated by native PAGE, and were confirmed to adopt a channel-like architecture using TEM. However, heterogeneity prevented determination of precise stoichiometry by rotational averaging. Nevertheless, M complexes mediated membrane bilayer permeability using an indirect ion channel assay, based upon fluorescent

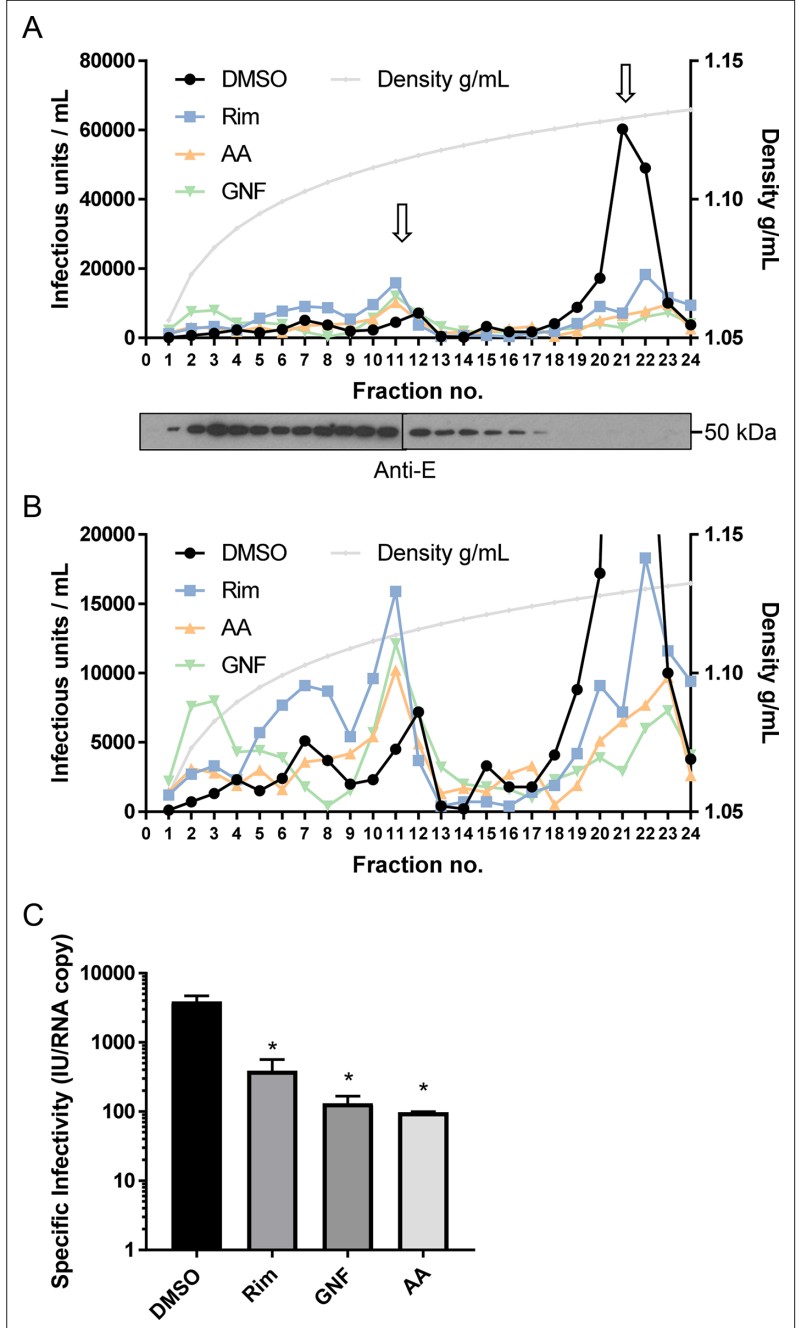

**Figure 7.** M channel inhibitors lack virus-lytic or cell-targeted effects. To exclude the possibility that early effects upon ZIKV infection were due to a directly damaging effect of compounds upon virion integrity, particles were concentrated and purified from Vero cell supernatants by PEG-precipitation and ultracentrifugation through a sucrose cushion. Virions were then dosed with inhibitors prior to ultracentrifugation through a continuous iodixinol gradient. This not only purifies virions according to buoyant density, but separates them from small molecules, precluding off-target effects upon cells during ensuing infection (see methods for details). (**A**) Infectivity profile from a representative gradient experiment where purified virions were treated with rimantadine (40μM), GNF 5837 (1 μM), or AA 29504 (10 μM) prior to ultracentrifugation (or, DMSO at equivalent concentrations). 10μl from each fraction was then used to infect naive Vero cells in a 96-well plate, representing a 10-fold dilution. Infected cells were detected by immunofluorescence and counted as described elsewhere. A further 15μl was analysed by western blotting for the (E)nvglycoprotein (lower panel). (**B**) Zoomed in view of gradient in A, with Y-axis capped at 20,000 units to delineate infectivity profiles. (**C**) Specific infectivity of virions in the presence/absence of inhibitors

*Figure 7 continued on next page*

*Figure 7 continued*

based upon q-RT-PCR normalisation confirms the reduction of infectivity rather than disrupted virions. p≤0.05, Student's t-test. Error bars represent standard deviations based upon n=3 biological replicates.

The online version of this article includes the following source data and figure supplement(s) for figure 7:

**Source data 1.** Infectivity data and fraction density (normalised) for iodixinol gradients with untreated or inhibitor treated purified ZIKV particles, titred on Vero cells by fluorescence focus forming assay.

**Figure supplement 1.** Effect of dilution upon measured titre for ZIKV purified through iodixinol gradients, as described in *Figure 7*.

**Figure supplement 1—source data 1.** Prism file with infectivity data for serially diluted gradient fractions containing purified ZIKV particles, titred on Vero cells.

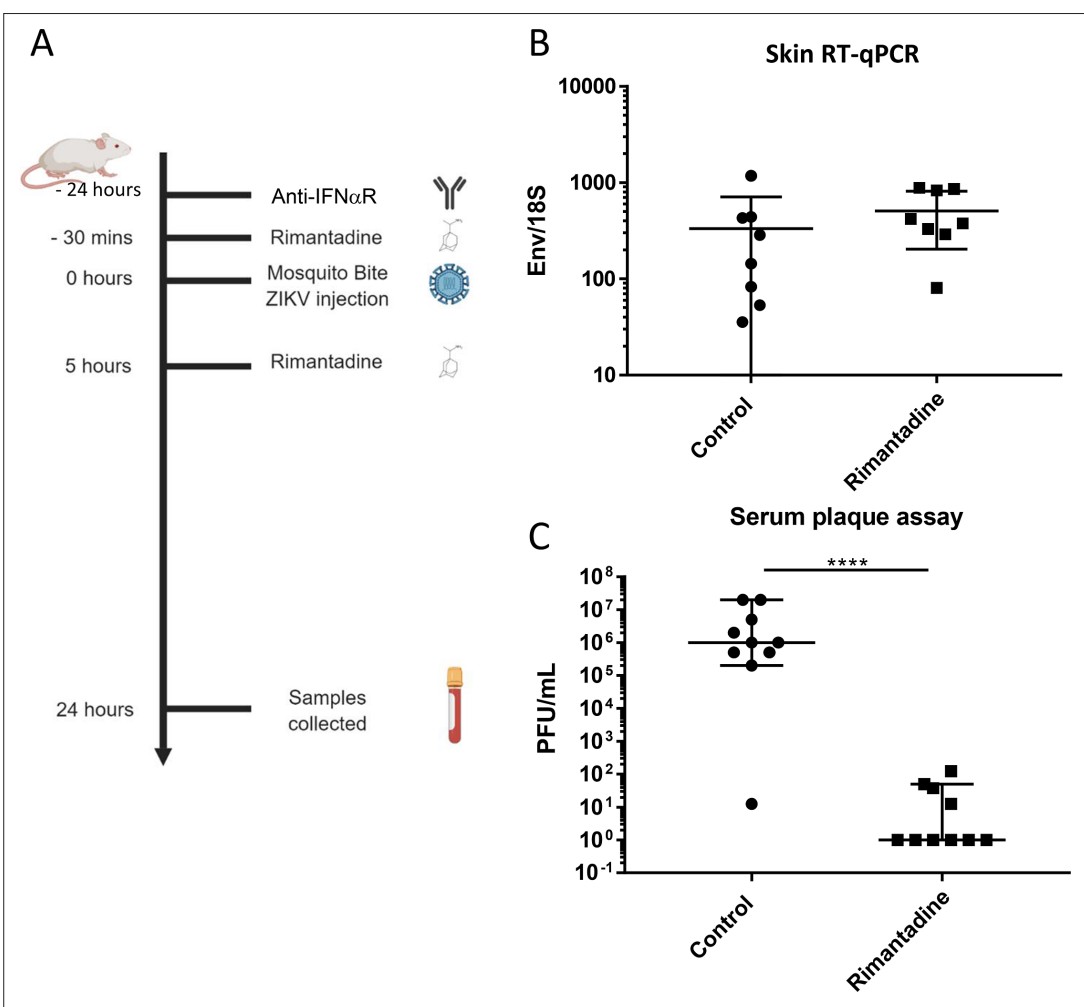

**Figure 8.** Anti-ZIKV activity in preclinical models. (**A**) Schematic of preclinical experiments involving transient IFNAR blockade, augmentation of ZIKV infection through mosquito bites and treatment with rimantadine. (**B**) RT-qPCR for ZIKV E relative to 18S RNA from skin tissue at injection site derived from one of two representative experiments. (**C**) Infectious titre of serum derived from rimantadine treated or control mice, determined by plaque assay in BHK21 cells. ****p ≤ 0.0001, Student's *t*-test.

The online version of this article includes the following source data for figure 8:

**Source data 1.** Infectivity and RT-qPCR data for serum and skin, respectively, for mice infected with ZIKV alongside transient blockade of IFNAR1 and mosquito bites, with or without rimantadine treatment as per schematic.

dye release from liposomes. Dose-dependent dye release was not only increased by acidic pH, but was also sensitive to rimantadine. Importantly, rimantadine sensitivity supports that dye release was mediated by interactions with a folded protein complex rather than via aggregates disrupting membrane integrity, and increased activity at acidic pH is consistent with activation within acidifying endosomes during virus entry. Other pH-gated channels including IAV M2, HCV p7 (genotype and polymorphism dependent) and HPV E5, are similarly activated within the same assay system (*Wetherill et al., 2012*; *Scott et al., 2020*; *Shaw et al., 2020*; *Atkins et al., 2014*). However, the most obviously solvent-accessible, conserved, ionisable residue present within M protein peptides, His28, had effectively no influence upon pH activated activity. It will be of interest to understand which residues might mediate such effects, assuming they are recapitulated within cellular membranes.

Rimantadine was able to interrupt ZIKV infection of Vero cells, causing a dose-dependent reduction in the number of cells infected and ensuing virus protein expression. The $EC_{50}$ for rimantadine was between 10 and 20 μM, which is comparable to drug potency versus susceptible HCV strains (*Griffin et al., 2008*). Rimantadine was most effective when added prior to and simultaneously with infectious innoculae, suggesting that pre-loading of endocytic vesicles with the drug may expedite the inhibition of virion-resident channels. Interestingly, adding rimantadine both during and after infection was consistently less effective than adding it during the infection alone. We are uncertain why this should be the case, yet we speculate that in addition to inhibitory properties, rimantadine may also exert a drug rescue effect upon cells during multi-cycle replication, lessening the cytopathic effects of ZIKV infection and thereby artificially inflating the number of surviving cells remaining at the end of the assay.

However, rimantadine also potently suppressed ZIKV replication in a preclinical model, supporting that M channels constitute a physiologically relevant drug target. Rimantadine treatment prevented viraemia, consistent with an effect upon virus spread linked to drug effects observed in culture. To date only M2, and more recently SARS-CoV-2 E, are the only examples where targeting a viroporin has been shown to exert preclinical antiviral efficacy. Thus, our findings strongly support that M channels are also physiologically relevant drug targets.

In lieu of structural information for channel complexes, we employed molecular modelling and MD simulations to understand how channels might be formed from either dimeric, or monomeric M protein. Importantly, monomer simulations indicated that a single *trans*-membrane-spanning topology was unstable resulting in the C-terminus folding back towards the plane of the membrane surface. Hence, we considered that dual-*trans*-membrane topology protomers comprise the more energetically favourable conformation for mature M, as seen for multiple structures of mature *Flavivirus* virions (*Sirohi et al., 2016*). This was despite some prediction packages including TMHMM, TOPCONS, and Phobius, supporting that M formed single-pass protomers. In contrast, MEMSAT-SVM not only predicted dual-pass conformations, but also predicted that helix 3 would most likely be pore lining, despite the majority of lumenal residues being hydrophobic in nature.

Compact hexameric models for M channel complexes comprised of dual-spanning protomers displayed consistent channel-like characteristics compared to an array of alternative conformations and oligomeric states, with helix 1 contributing significantly to overall channel stability. Channel closure occurred in proximity to Leu64, closing off the water column.

Compact hexamer channel models have considerable limitations, despite performing best in MD analyses. The omission of the amino-terminal 17 amino acids from simulations was to recapitulate the sequence of the synthetic peptides, plus it is predicted to be intrinsically disordered. Nevertheless, disordered protein regions are increasingly implicated in various protein functions and may influence other roles of the M protein. We are developing in-house recombinant expression systems to investigate full-length M, modifying simulations accordingly in parallel. Another limitation of current channel models is uncertainty around protomer stoichiometry, topology, and orientation, although available data support that dual-transmembrane conformations are the most stable. However, they provided a seemingly robust template for the in silico screening of small molecules with improved potency compared to prototypic channel blockers. It will be important going forward to investigate the potential formation of oligomeric complexes within virion and/or cellular membranes to further validate a role for M channels, and to complement the functional studies based upon small molecule inhibitors described herein.

**Table 3.** Properties of hits taken forward into ZIKV cell culture screens, including effects versus canonical targets.

| Drug | M-binding site (EC/IC$_{50}$) | Canonical target (Ki/EC/IC$_{50}$) | MTT effect? |
|---|---|---|---|
| Rimantadine | Lumen (80µM) | IAV M2 (variable) | No |
| GNF 5837 | Lumen (~80nM) | Tropomyosin receptor kinases (TrK) A, B, and C (10nM) | No |
| KB SRC 4 | Lumen (~250nM) | c-Src (Ki = 44nM) | Yes, 500nM |
| GSK 2837808A | Lumen (~40µM) | Lactate dehydrogenase A and B (LDHA/B), 2.6/43nM | No |
| Formoterol | Periphery (~80µM) | β$_2$-Adrenergic receptor agonist (pM range) | Yes, 100µM |
| L-732, 183 | Periphery (~40µM) | Tachykinin NK$_1$ receptor antagonist (IC$_{50}$ = 2.3nM) | No |
| AA 29504 | Periphery (~10µM) | GABA$_A$ receptor agonist (IC$_{50}$ = 9–13µM) | Marginal, 20µM |

Whilst a blunt tool in terms of potency, rimantadine (or other prototypic blockers) binding can signify druggable binding sites within viroporin complexes (*Foster et al., 2014*; *Scott et al., 2020*; *Shaw et al., 2020*). Rimantadine was predicted to bind within a lumenal cavity proximal to Leu64, repeated along the sixfold radial symmetry of the channel model. In addition, the external membrane-exposed face of the same part of the channel model also contained a cavity predicted to be conducive to binding small molecules. The presence of two binding sites for small molecules could enable M-targeted drug combinations that might minimise the chance of resistant variants being selected, potentially also achieving synergistic antiviral effects. In agreement, we found that two planar 'JK series' molecules from previous work on HCV p7 exhibited activity versus M channels in vitro, consistent with the shape composition of the P1 site (*Figure 6—figure supplement 1*).

Thus, a series of chemically distinct molecules was identified using in silico HTS targeting each binding site, derived from a drug repurposing library. Hits were shortlisted for predicted binding accounting for hydrophobic exposure-induced penalties at the peripheral site, chemical properties (Lipinski), and those predicted to bind both sites were removed from both lists. Screening of short-listed compounds using dye release assays identified multiple hits targeting both sites, several of which were corroborated by testing for antiviral effects in ZIKV infectious assays. (*Table 3*).

Despite the correlation of anti-M potencies in vitro with antiviral effects, it cannot yet be excluded that antiviral efficacy may result from indirect mechanisms linked to the canonical cellular targets of these repurposed ligands. By far the most potent hit against M was GNF 5837, which was predicted to bind the L1 site. GNF 5837 is a potent inhibitor of TrKs A, B, and C, high-affinity receptors for nerve growth factor that regulate neuronal survival and differentiation (*Albaugh et al., 2012*). GNF 5837 has an IC$_{50}$ of ~10 nM against its cognate targets, yet only achieved an antiviral ~EC$_{50}$ at 80 nM in BHK cell plaque-reduction assays. Importantly, TrK expression is primarily restricted to the (central nervous system) CNS and the thyroid, with only very low expression in kidney tissues according to the human protein atlas. Moreover, we could find no reference to TrK expression in either Vero or BHK cells. Other lumenal hits comprised the c-Src inhibitor, KB SRC 4 (*Brandvold et al., 2012*) (250 nM), and GSK 2837808A (*Billiard et al., 2013*) (40 µM), an inhibitor of lactate dehydrogenases A and B (LDHA/B). KB SRC 4 has an in vitro Ki of 44 nM versus c-Src (higher against other Src-related kinases), whilst GSK 2837808A has an IC$_{50}$ of 2.6 and 43 nM against LDHA/B, respectively; the former impeded cell metabolic activity at 500 nM, whilst the latter had no effect. It will be critical to establish, for example via use of knock-down cell lines, whether compounds such as GSK 2837808A, GNF 5837, and KB SRC 4 are indeed affecting ZIKV replication specifically, as it is eminently conceivable that their effects may be indirect.

The peripheral hits tested in culture comprised Formoterol hemifumarate (80 µM), L-732 183 (40 µM), and AA 29504 (10 µM). Only high concentrations of Formoterol impeded metabolic activity (100 µM). Formoterol is a long-acting β$_2$-adrenergic receptor agonist (IC$_{50}$ in pM range), used as a bronchodilator in the treatment of chronic obstructive pulmonary disease (*Berger and Nadel, 2008*). L-732 183 is a potent competitive tachykinin NK$_1$ receptor antagonist (IC$_{50}$ = 2.3 nM), the receptor for substance P (*Quartara and Maggi, 1997*), whilst AA 29504 (*Vardya et al., 2012*) is a positive allosteric modulator of GABA$_A$ receptors (IC$_{50}$ = 9–13 µM). Again, whilst there is a considerable difference in potency for both Formoterol and L-732 183 targeting their canonical targets compared to

antiviral effects, it will be critical to establish whether these are independent phenomena. Notably, this is less clear in terms of differential potency of AA 29504, although, much like the case for GNF 5837, GABA$_A$ receptors, the target of AA 29504, are not expressed in the kidney. However, it should be emphasised that repurposed compounds provide the structural basis for further bespoke drug development targeting M channels, rather than being used directly as antivirals, providing opportunities to further ensure specificity through the establishment of structure–activity relationships (SARs) and the selection of drug resistant ZIKV.

Further evidence that repurposed compounds and rimantadine exerted specific antiviral effects resulted from density gradient separation of ZIKV virions from unbound compounds, as described previously (*Shaw et al., 2020*). This ensures that target cells are highly unlikely to be exposed to meaningful concentrations of each compound, minimising one possible source of off-target effects. Moreover, virolytic mechanisms were found to be similarly unlikely to explain compound effects as the specific infectivity of virions was diminished by drug treatment, normalised by ZIKV RNA content. Interestingly, compounds specifically reduced infectivity in the major high-density peak, but not in a lower-density region of the gradient where lower levels of infectivity were susceptible to relatively low serial dilutions; this was not the case for the high-density peak. Interestingly, the only published study we identified relating to multiple particle types in *Flaviviruses* was a study assessing the effects of pr-M cleavage site mutations in the morphogenesis and secretion of tick-borne encephalitis virus subviral particles (SVPs), generated via pr-M-E expression alone (*Allison et al., 2003*). Preventing pr-M cleavage by furin increased the proportion of larger, faster sedimenting SVPs corresponding in size to virions, and which harboured altered E protein glycosylation. Whilst the link between altered particle species, M processing and, potentially, M channel function is intriguing, it currently remains unclear whether these aspects are linked mechanistically.

The compounds identified herein represent only the first steps towards the design of bespoke M inhibitors, which could be enhanced through the exploration of chemical space around potent repurposed ligands; ideally, this will enable the continued emphasis on developing small molecules with superior qualities compared to rimantadine, or other prototypic viroporin inhibitors. Nevertheless, given its activity in vivo and in cell culture, there may be value in considering the immediate clinical repurposing of rimantadine to combat severe ZIKV infection and/or *trans*-placental transmission via early-stage clinical trials. The epidemic in South America and ensuing worldwide spread illustrated the speed with which more pathogenic ZIKV can be transmitted, and outbreaks continue in Africa and the Indian subcontinent. ZIKV is also unique amongst *Flaviviruses* as it can also spread from person to person via sexual contact (*D'Ortenzio et al., 2016*), as well as tragically infecting the unborn foetus causing microcephaly (*York, 2021*).

It is possible that emerging ZIKV replicase-targeted drugs such as Galidesivir will not be safe to use during pregnancy due to potential teratogenic effects, although recent studies in Rhesus macaques did not indicate any issues (*Lim et al., 2020*). The FDA classes rimantadine as a class C drug during pregnancy, that is administered if the potential benefit might outweigh the risk. Whilst rimantadine will cross mouse placentas, eleven times the maximum recommended human dose (MRHD) is necessary to become embryotoxic. However, 3.4 and 6.8 times the MRHD did increase pup mortality up to 4 days post-partum, yet this still by far exceeds the dose used during influenza prophylaxis in the past (Product Information. Flumadine (rimantadine). Forest Pharmaceuticals, St. Louis, MO. 2007). No controlled studies have been undertaken during human pregnancy, but faced with the alternative of microcephaly, the balance of risk may be favourable. The generic status of rimantadine also means that it could be deployed rapidly at minimal cost to areas of endemic ZIKV infection or during outbreaks, which overwhelmingly tend to occur in lower/middle income countries (LMICs).

In summary, this work supports the formation of membrane channels by the ZIKV M protein, consistent with viroporin activity. Blocking this activity with small molecules reveals an important role during the early stages of ZIKV replication, consistent with virus entry and/or uncoating, and this also translates to antiviral efficacy a preclinical model. Rimantadine may constitute an easy-to-access generic drug for the treatment and/or prophylaxis against ZIKV in LMIC. Moreover, the selection of new repurposed ligands with anti-M activity demonstrates potential for developing novel therapies targeting this protein function, which in turn may also apply to other *Flaviviruses*. In addition, the relative enrichment of effective ligands versus both a lumenal and a peripheral binding site, combined with improved potency compared to rimantadine, further demonstrates the value of molecular models as

templates for future drug development. Confidence in such models will be further enhanced via iterative drug development to generate an SAR for inhibitors targeting each site. In the longer term, structural information on M channel structures, ideally located within virion membranes, will be essential to both developing novel therapeutics, and then in turn, using these to investigate the precise roles for M channel function during ZIKV infection.

## Materials and methods

### Cell culture

Vero (African Green Monkey kidney (ATCC Cat# CRL-1586, RRID:CVCL_0574)) and BHK-21 (baby hamster kidney (ATCC Cat# CCL-10, RRID:CVCL_1915)) cells were cultured in Dulbecco's modified essential cell culture media, supplemented with 10% (foetal calf serum) FCS and 100 units/ml penicillin and 0.1 mg/ml streptomycin, at 37°C in 5% $CO_2$ in a humidified culture incubator. Cells were passaged every 2–3 days using trypsin/(Ethylenediaminetetraacetic Acid) EDTA (Sigma), sub-dividing cultures using ratios between 1:5 and 1:10, depending upon confluency. C6/36 cells (derived from *A. albopictus* mosquitoes (ATCC Cat# CRL-1660, RRID:CVCL_Z230)) were cultured in L-15 media, supplemented with 10% (Triptose phosphate broth) TPB, 10% FCS and 100 units/ml penicillin and 0.1 mg/ml streptomycin, at 28°C with no added $CO_2$.

### Virus stocks

ZIKV/*H. sapiens*/Brazil/PE243/2015 (PE243) Zika virus (**Donald et al., 2016**) was obtained from a patient in Recife. It has been sequenced and was supplied (AK laboratory) as a frozen viral stock. This was grown once in BHK-21 cells then passaged once in C6/36 cells, titrated, and stored frozen (−80°C) at $6 \times 10^6$ PFU/ml. New stocks for use in cell culture were generated in house at $1.6 \times 10^6$ PFU/ml in Vero cells (cell culture assays). C6/36-derived stocks were used for in vivo assays. Briefly, ~$6 \times 10^6$ Vero cells were seeded into a T75 (Corning) and left to settle over 4 hr. Cells were then washed once in (phosphate buffered saline) PBS, prior to addition of PE243 virus in complete (Dulbecco's Modified Eagle Medium) DMEM media + 10 mM (4-(2-Hydroxyethyl)piperazine-1-ethane-sulfonic acid) HEPES (Gibco), at a multiplicity of infection (MOI) of 0.001 PFU/cell. Infections were allowed to proceed for 1 hr at 37°C, 5% $CO_2$, then supernatants were replaced with fresh media. Development of ~60% cytopathic effect led to harvesting and clarification of supernatants ($3184 \times g$, 20 min, 4°C, in an Eppendorf 5810 R centrifuge). Aliquots were then titred by plaque assay (see below), snap frozen, and stored at −80°C.

### Determination of virus titre and inhibitor assays

Plaque assays for titration (Vero) or plaque-reduction (BHK21) assays were performed upon cells at 80% confluency in 12-well plates. For titration, 10-fold serial dilutions of viral supernatants were made in 0.75% PBSA (PBS containing 0.75% bovine serum albumin) and 200 μl added to wells for 1 hr with rocking every 15 min. 2 ml overlay media were then added (2× (Minimal Essential Media) MEM medium (Gibco), 4% FCS (Gibco), 200 units/ml penicillin and 0.2 mg/ml streptomycin, mixed with viscous 1.2% Avicel (FMC Biopolymer)) and cells incubated for 5 (Vero) or 3 (BHK21) days at 5% $CO_2$ and 37°C. Supernatants were removed and cells fixed in PBS/10% paraformaldehyde (PFA) for 1 hr at 4°C prior to staining with 0.1% Toludine Blue (Sigma) for 30 min. Plaque-reduction assays comprised duplicate wells per condition with 10 PFU added in the presence of inhibitor or DMSO solvent control (maximum of 0.1% vol/vol).

Single cell infectious assays were conducted upon Vero cells seeded at 2000 cells per well in a 96-well cell culture dish (Greiner Bio-One). After settling for 4 hr, cells were incubated for 1 hr with virus stock diluted in complete medium at an MOI of 1 PFU/cell. Following infection virus-containing media was removed, cells were washed with PBS and replaced with fresh cell culture media, prior to incubating at 37°C and 5% $CO_2$ for 48 hr in a humidified incubator. Cells were then washed three times in PBS, fixed using 4% PFA for 10 min at RT, and stained for ZIKV E. Briefly, cells were permeabilised using 0.1% TX-100 in PBS at (Room temperature) RT for 10 min, then stained using a mouse monoclonal anti-ZIKV E antibody (Aalto Bio Reagents #AZ1176) diluted 1:500 in 10% vol/vol FCS in PBS overnight at 4°C. Cells were washed three times in PBS prior to adding goat anti-mouse Alexa Fluor 488-nm-conjugated secondary antibody (Thermo Fisher Scientific Cat# A-11001, RRID:AB_2534069),

diluted 1:500 in 10% vol/vol FCS in PBS at RT for 1 hr. Cells were washed another three times in PBS and left in PBS for imaging. Infectious units (FFU) were determined using an IncuCyte Zoom (Essen Bioscience) to determine numbers of fluorescently labelled cells as well as confluency, as described previously (*Stewart et al., 2015*). The ×10 objective was used to take four images covering each well, with positive and negative control wells allowing modification of the processing definition parameters for optimal detection.

## Preclinical ZIKV model

In vivo animal models were approved by the University of Leeds local ethics review committee. Procedures were carried out in accordance with the United Kingdom Home Office regulations under the authority of the appropriate project and personal license (awarded to CSM, and CSM/DL, respectively). Wild-type C57BL/6j mice (RRID:IMSR_JAX:000664) were bred in-house and maintained under specific pathogen-free conditions. Mice were age and sex matched in all in vivo experiments, and used between 4 and 12 weeks of age.

Mice were dosed with 1.5 mg InVivoMAb anti-mouse IFNAR-1 antibody (Sino Biological Cat# 40543-R033, RRID:AB_2860467) 24 hr prior to virus inoculation, then a sub-cutaneous dose of rimantadine hydrochloride at 20 mg/kg (or carrier control) 30 min prior to ZIKV infection. Mice were then anaesthetised using 0.1 ml/10 g of Sedator/Ketavet via intraperitoneal injection, then placed on top of mosquito cages using foil to expose only the dorsal side of one hind foot. No more than 5 mosquitoes were allowed to feed on each mouse. 2000 PFU of C6/36-derived ZIKV was injected directly into the bite site using a 5 µl 75 N syringe, 26ga (Hamilton) using small RN ga33/25 mm needles (Hamilton). A second bolus of rimantadine was administered 5 hr post-infection. Mice were observed four times throughout the 24 hr experiment, weighed once, and culled 24 hr post-infection.

Mice were culled via a schedule 1 method. Skin from the bitten foot was dissected and placed in 0.5 ml RNAlater (Sigma-Aldrich, USA) in 1.5 ml tubes. Blood was collected from the ventricles by cardiac puncture, and then centrifuged to isolate serum and stored at −80°C.

## Mosquito rearing

*A. aegypti* (Liverpool strain, RRID:NCBITaxon_7159) mosquitoes were reared at 28°C with 80% humidity conditions with a 12-hr light/dark cycle. Eggs were hatched overnight from filter papers in trays containing approximately 1.5 cm depth of water. Larvae were fed Go-cat cat food until pupation. Pupae were placed in water-filled containers inside BugDorm mosquito cages where they were left to emerge. A 10% wt/vol sucrose solution was fed to adult mosquitoes. Mosquitoes were ready for biting experiments from 21 days post-hatching.

## RNA purification and quantification

Skin tissue was lysed in 1 ml Trizol (Invitrogen/Thermo) by shaking with 7 mm stainless steel beads at 50 Hz for 10 min on a Tissue Lyser. 200 µl chloroform was added, and samples mixed by inversion, prior to centrifugation in a microcentrifuge at 12,000 × $g$ for 15 min at 4°C and removal of the top aqueous phase to a fresh tube containing an equal volume of 70% (vol/vol) EtOH. RNA was then extracted using a PureLink RNA Mini kit (Life Technologies) according to the manufacturer's instructions. cDNA was synthesised from 1 µg of RNA using the 'Applied Biosystems High-Capacity RNA to cDNA' kit, according to the manufacturer's instructions.

cDNA was diluted 1 in 5 in RNAse-free water, then introduced into a master mix comprising cDNA, primers (ZIKV E Fwd: AGGCAAACTGTCGTGGTTCT; ZIKV E Rev: TCAGACCCAACCACATCAGC), water and SYBRÒ green mix. A non-template control comprising RNAse-free water in place of cDNA was also included. Triplicate technical replicates were made for each biological replicate and a standard curve generated by 10-fold serial dilution of the $10^{-2}$ PCR standard. (polymerase chain reaction) PCR plates were run on an Applied Biosystems Quantstudio 7 flex machine. Ct value was calculated automatically using the Quantstudio software, which detects the logarithmic phase of the PCR reaction. Samples were quantified according to their position on the standard curve, which was required to have close to 100% efficiency, indicated by $R^2 \geq 0.998$ and a slope of 3.3. Melt curves were conducted to control for primer specificity. Analysis of qPCR data was done with Microsoft Excel by the use of the median of the technical replicates and normalising them to the median of the technical replicates

of the housekeeping genes. GraphPad Prism software was used to generate graphs and the non-parametric Mann–Whitney test was used for comparisons between two groups.

## Western blotting

6-well plates seeded with $1 \times 10^5$ Vero cells were infected with ZIKV at an MOI of 0.1 PFU/cell as above, then media applied containing increasing concentrations of rimantadine or a DMSO solvent control (0.1% vol/vol). Cells were harvested at 48 hr post-infection by washing three times in 1 ml PBS and scraping into a 1.5-ml Eppendorf tube. Cells were pelleted at $10,000 \times g$ in a microcentrifuge, then whole-cell lysates were made in 200 µl Enriched Broth Culture lysis buffer [50 mM Tris HCl, pH 8.0, 140 mM NaCl, 100 mM NaF, 200 µM $Na_3VO_4$, 0.1% (vol/vol) sodium dodecyl sulfate (SDS), 1% Triton X-100, Roche complete ULTRA protease inhibitor cocktail]. Lysates were normalised for protein concentration using a Pierce BCA Protein Assay Kit (Thermo Fisher Scientific), then diluted with an equal volume of 2× Laemmli Buffer (100 mM Tris HCl pH 6.8, 4% (vol/vol) SDS, 20% (vol/vol) glycerol, 10 mM (dithiothreitol) DTT, 0.025% (wt/vol) Bromophenol Blue). Lysates were denatured by heating for 10 min at 95°C prior to separation on hand-cast Tris-glycine polyacrylamide gels. Proteins were resolved by SDS–PAGE (25 mM Tris–Cl, pH 8.0, 250 mM glycine, 0.1% SDS) then transferred to a polyvinylide fluoride (Immunoblot-FL Merck Millipore) membrane, pre-activated in 100% MeOH, using a Hoeffer semi-dry transfer rig, sandwiched in blotting paper soaked in transfer buffer (25 mM Tris–Cl, 250 mM glycine, pH 8.3, 20% vol/vol MeOH). Transfers proceeded for 1–2 hr at 120–240 mA constant current, depending on the number of gels. Membranes were blocked in 5% wt/vol fat-free milk in TBS-T (Tris-buffered saline (50 mM Tris–Cl, pH 7.5, 150 mM NaCl) with 0.1% vol/vol Tween 20, Sigma-Aldrich) for 4 hr at RT with gentle shaking, prior to incubation with primary antibodies diluted in 5% (wt/vol) BSA in TBS-T at 4°C overnight with gentle shaking (Anti-E 1:10,000, mouse monoclonal, Aalto Bio Reagents #AZ1176). Membranes were then washed three times for 10 min in TBS-T, prior to incubation with secondary antibody (1/10,000 goat anti-mouse IgG–horseradish peroxidase conjugate, Sigma #A4416, RRID:AB_258167). Blots were visualised using ECL prime western blotting detection reagent (GE Healthcare Life Sciences) by exposure to X-ray film, with protein sizes determined by comparison with pre-stained molecular weight markers (Seeblue Plus2, Invitrogen).

## Purification and density gradient separation of ZIKV virions

This was undertaken using similar methods to our previous studies of HCV (*Shaw et al., 2020*). Briefly, 10 ml of infectious ZIKV culture Vero cell supernatants were clarified and concentrated by the addition of PEG-8000/PBS to a final concentration of 10% wt/vol, mixing by several inversions, and incubation overnight at 4°C. Virion concentrates were pelleted at 2000 rpm in a benchtop centrifuge at 4°C and then resuspended in 0.5 ml PBS. This was then layered over a 1-ml 20% wt/vol in PBS sucrose cushion and virions pelleted at 100 K × $g$ in a Beckmann TLS-55 rotor at 4°C for three hr. Pellets were left to resuspend for 1 hr at 4°C in 200 µl PBS, then layered over a 10–40% vol/vol iodixinol/PBS gradient. These were spun in the same conditions as above but for 2 rather than 3 hr. Gradients were fractionated from the top into 24 equal samples by careful pipetting, then 10 µl diluted into media to assess infectivity in Vero cells seeded into 96-well plates as described above. Virion RNA was harvested from samples and quantified as described above. Western blotting was performed upon 15 µl of gradient sample with 5 µl 4× Laemmli sample buffer added followed by boiling for 10 min prior to SDS–PAGE and immunoblotting for E protein as described above.

## Native PAGE

M peptide was solubilised at 37°C for 10 min in 300 mM detergent 1,2-dihexanoyl-sn-glycero-3-phosphocholine (DH$_6$PC), in Liposome Assay Buffer (10 mM HEPES, pH 7.4, 107 mM NaCl). Native-PAGE loading dye (150 mM Tris–Cl pH 7.0, 30% (vol/vol) glycerol, 0.05% (wt/vol) bromophenol blue) was added to samples, which were loaded onto gradient precast gels (4–20%) (TGX, Bio-Rad) and run using Native-PAGE running buffer (250 mM Tris–Cl, pH 8.5, 192 mM glycine) at 140 V for 1 hr. Gels were stained with Coomassie Brilliant Blue (0.1% (wt/vol) Coomassie Blue, 10% acetic acid, 50% MeOH). Unstained SDS-free molecular weight markers (Sigma-Aldrich) were used to estimate protein size.

## Synthetic M peptide

M peptide with an N-terminal truncation (residues 18–75) was manufactured by Alta Bioscience, provided as a lyophilised powder at >95% purity based upon HPLC (*Figure 1—figure supplement 1*). Peptides contained the sequence:

Ac – ESREYTK(H/A)LIKVENWIFRNPGFALVAVAIAWLLGSSTSQKVIYLVMILLIAPAYS

## Transmission electron microscopy

5 μg of M peptide was incubated in 10 mM HEPES, 107 mM NaCl and varying concentrations of $DH_6PC$ for 10 min at RT. Sample was added to copper grids with a continuous amorphous carbon film, before washing with water, then stained with 1% uranyl acetate. Grids were examined using a Tecnai F20 at 120 kV on a FEI CETA camera with a sampling of 4.18 Å per pixel. Particles were manually picked and 2D class averaging were carried out using RELION 3 software (*Zivanov et al., 2018*).

## Liposome dye release assay

L-α-phosphatidic acid (α-PA), L-α-phosphatidyl choline (α-PC), and L-α-phosphatidyl ethanolamine with lissamine rhodamine b labelled headgroups (α-PE) from chicken eggs, supplied at 10 mg/ml in chloroform, were purchased from Avanti Polar Lipids (Cat# 840101, 131601, 810146). Aliquots were made using Hamilton glass syringes in glass vials, then stored at −80°C. Lipids were combined into a glass tube on ice under non-oxygen gas (Nitrogen), as a total of 1 mg (50 μl PA, 50 μl PC, and 5 μl PE). Lipids were dried initially under nitrogen, then overnight using a vacuum desiccator at RT. Lipids were then rehydrated in CF-containing buffer (50 mM CF, 10 mM HEPES–NaOH pH 7.4, 107 mM NaCl) to 2 mg/ml, with vigorous shaking overnight at RT. The following day, liposomes were generated via extrusion (15 passes) using an Avanti mini extruder housing a 0.4-μM filter (Whatmann), at 37°C. Resulting unilamellar liposomes were washed at least three times in liposome assay buffer (10 mM HEPES pH 7.4, 107 mM NaCl) to remove unincorporated CF, pelleting liposomes at 100,000 × *g* for 15 min at RT using a MLS-50 rotor in a Beckman Coulter TLX ultracentrifuge. Final pellets were resuspended in assay buffer (500 μl) and concentration determined by comparing rhodamine absorbance ($OD_{570\,nm}$) in pre-extrusion (diluted 1:20) and final samples:

$$\text{Liposome concentration (mM)} = \left\{ 2.75\,\text{mM (average lipid molarity} / (\text{OD570 pre} \times 20) \right\} \times \text{OD570 post}$$

Dye release assays were conducted using up to 800 nM peptide (480 nM standard) in DMSO (maximum of 5% vol/vol DMSO per well), with 50 μM liposomes in a total reaction volume of 100 μl. Inhibitors were pre-incubated with peptide for 5 min in flat-bottomed, black 96-well plates (Greiner Bio-One) at RT, prior to the rapid addition of chilled liposome suspensions in assay buffer. CF release compared to solvent controls and liposomes alone was measured over 30 min at 37°C with initial mixing, using a FLUOstar Galaxy Optima plate-reader BMG Labtech, $\lambda_{ex}$ 485 nm/$\lambda_{em}$ 520 nm. Gain adjustment at 90% total fluorescence was set using a pre-measured sample containing 0.5% vol/vol Triton X-100 to lyse all liposomes present. Endpoint values from three technical repeats were calculated for each averaged biological repeat and significance between the latter determined using multiple paired Student's *t*-tests. Assays assessing pH dependence utilised assay buffers at stated pH, with endpoint supernatants re-buffered using 2.5 μl 1 M Tris–Cl [pH 7.5] prior to clarifying by ultracentrifugation as above and determining fluorescence.

## In silico analysis of M protein

Alignment of *Flavivirus* M sequences was performed using Clustal Omega (https://www.ebi.ac.uk/Tools/msa/clustalo/, RRID:SCR_001591), with outputs visualised using BOXSHADE 3.21 (https://embnet.vital-it.ch/software/BOX_form.html, RRID:SCR_007165), and curated in Jalview (RRID:SCR_006459). Conservation outputs were fed into an online tool WEBLOGO (https://weblogo.berkeley.edu/, RRID:SCR_010236) to generate a protein sequence LOGO to illustrate the high degree of conservation between >700 isolates.

M protein tertiary structures were taken from PDB: 5IRE. Online servers used to determine probability of *trans*-membrane regions comprised TMHMM v2.0 (http://www.cbs.dtu.dk/services/TMHMM/, RRID:SCR_014935), TOPCONS (https://topcons.cbr.su.se/, RRID:SCR_006977), Phobius

**Table 4.** Summary of course-grain simulations.

| CG simulation | Membrane composition | Duration (µs) |
|---|---|---|
| 1 TMD POPC | POPC (100) | 5 × 3 |
| 2 TMD POPC | POPC (100) | 5 × 3 |

(https://phobius.sbc.su.se/, RRID:SCR_015643), SPLIT v4.0 (http://splitbioinf.pmfst.hr/split/4/), and MEMSAT-SVM (https://bio.tools/memsat-svm, RRID:SCR_010248). Amino acid properties were determined using EMBOSS Pepinfo (https://www.ebi.ac.uk/Tools/seqstats/emboss_pepinfo/, RRID:SCR_008493).

## Coarse-grained MD simulations

Raw data for all simulations can be found at https://doi.org/10.5518/1505.

Coarse-grained (CG) Molecular Dynamic simulations of monomeric M proteins were performed using Martini v2.2 force field (*de Jong et al., 2013*) (RRID:SCR_021951) and GROMACS (*Van Der Spoel et al., 2005*) (RRID:SCR_014565). The cryo-EM structure of ZIKV M protein structure (PDB: 5IRE) (*Sirohi et al., 2016*) was converted to a CG resolution using the 'martinize' script (RRID:SCR_022318). An elastic network model was used in the monomer simulations to maintain the secondary structure, yet not the tertiary structure. The cryo-EM structure was used to simulate the TMD model, whereas 'Modeller' (*Fiser and Sali, 2003*) (RRID:SCR_008395) was used to generate the second model, straightening the linker region between the two TMDs into a longer TMD. Five repeat simulations of 3 µs each were run for each system.

A POPC bilayer was built using INSANE (INSert membrANE) CG tool (*Wassenaar et al., 2015*). Systems were solvated with CG water particles and ions were added to neutralise the system to a final concentration of 150 mM NaCl. Prior to simulation, systems were energy minimised using the steepest descent algorithm for 500 steps in GROMACS and equilibrated for 10 ns with the protein back-bone restrained. The temperature was set at 323 K and controlled by V-rescale thermostat (coupling

**Table 5.** Summary of all-atom simulations.

| AA simulation | Membrane composition | Duration (ns) |
|---|---|---|
| **Hexamer** (lumenal helix; compact vs radial; pH) | | |
| H2 Radial | POPC (100) | 2 × 200 |
| H2 Radial protonated | POPC (100) | 2 × 200 |
| H2 Compact | POPC (100) | 2 × 200 |
| H2 Compact protonated | POPC (100) | 2 × 200 |
| H3 Radial | POPC (100) | 3 × 200 |
| H3 Radial protonated | POPC (100) | 3 × 200 |
| H3 Compact | POPC (100) | 3 × 200 |
| H3 Compact protonated | POPC (100) | 3 × 200 |
| **Heptamer** (lumenal helix; compact vs radial; pH) | | |
| H2 Radial | POPC (100) | 2 × 200 |
| H2 Radial protonated | POPC (100) | 2 × 200 |
| H2 Compact | POPC (100) | 2 × 200 |
| H2 Compact protonated | POPC (100) | 2 × 200 |
| H3 Radial | POPC (100) | 2 × 200 |
| H3 Radial protonated | POPC (100) | 2 × 200 |
| H3 Compact | POPC (100) | 2 × 200 |
| H3 Compact protonated | POPC (100) | 2 × 200 |

constant of 1.0) (**Bussi et al., 2007**). Pressure was controlled by Parrinello–Rahman barostat (coupling constant of 1.0 and a reference pressure of 1 bar) (**Parrinello and Rahman, 1981**). Integration step was 20 fs (**Table 4**).

## Atomistic MD simulations

The all-atom hexameric and heptameric M protein oligomers were first energy minimised prior to conversion into CG using the Martini 2.2 forcefield and as above inserted into the bilayer system using INSANE. The systems were then equilibrated in CG restraining the protein. The systems were then converted back into atomistic resolution using the Martini backward tool (**Wassenaar et al., 2014**). Simulations were then energy minimised, equilibrated for 20 ns with the protein Cα atoms restrained and run using the CHARMM36 (RRID:SCR_014892) force field (**Huang and MacKerell, 2013**). Two or three repeat simulations (see **Table 5** for more information) were run for 200 ns for each system. Temperature and pressure were controlled using the v-rescale thermostat (**Bussi et al., 2007**) and Parrinello–Rahman barostat (**Parrinello and Rahman, 1981**), respectively. Bond lengths were kept constant using the LINCS algorithm (**Hess et al., 1998**). The time-step was 2 fs and the temperature set to 323 K.

## Design of hexameric and heptameric M channel structures

A python script (RRID:SCR_008394) was used to calculate the co-ordinates of each monomer within the oligomeric structure, a radius of 1.3 nm was used for hexamer:

> n=6 # Number of sides; x=0 # Origin on x axis; y=0 # Origin on y axis: *r*=1.3 # Radius of Circle polygon is in for i in range (0, n): posx = x + (r * math.cos (2 * math.pi * i / n)); posy = y + (r * math.sin (2 * math.pi * i / n)) print ("% 2d: %2.4 f: %2.4 f" % (i, posx, posy))

## In silico docking and virtual HTS

Maestro (Schrödinger, RRID:SCR_016748) was used for assessing ligand interactions with the compact hexameric M channel structure. This was minimised in a lipid membrane environment using an Optimised Potentials for Liquid Simulations (OPLS) force field and used for docking of unbiased compound analogues. The Maestro SiteMap function was used to assess the druggability score for three potential lumenal binding sites (L1–3), and another on the membrane-exposed periphery was identified manually.

For unbiased screening, a library of 1280 FDA-approved molecules (TOCRIS Screen, Biotechne. https://www.ebi.ac.uk/Tools/seqstats/emboss_pepinfo/) underwent the LigPrep function using Maestro (Schrödinger) software. The in-built SiteMap function was then used on the M Protein structure to generate a Glide (RRID:SCR_000187) grid for the docking of Rimantadine in the lumenal site (under aqueous conditions) using extra precise (XP) Glide setting. This was followed by docking in the same site mimicking membrane conditions. Rimantadine was then docked into the peripheral site under both aqueous and membrane conditions to generate 32 poses in each of the four scenarios. The prepared Tocris library was also docked into the two sites on the M protein structure using the four scenarios and Glide settings established using Rimantadine (lumenal-aqueous, lumenal-membrane, peripheral-aqueous, and peripheral-membrane). Duplicate poses were removed, and a final short list of 50 molecules was generated at each for in vitro testing.

## Availability of materials

ZIKV M peptides can only be provided under an MTA if technical issues are shown to prevent commercial or in-house synthesis, within the limitations of stocks available in the SG laboratory. ZIKV M channel model PDB files are available upon request, again under an MTA. Raw data for MD simulations is available at https://doi.org/10.5518/1505.

## Acknowledgements

Schematics in **Figure 2A**, **Figure 3—figure supplement 1B** and **Figure 8A** were generated using Biorender software (https://biorender.com) and included here under an authorised CC-BY-NC-ND license. Simulations were undertaken on ARC3 and ARC4, part of the High-Performance Computing facilities at the University of Leeds. Work was supported by Institute PhD Scholarships from the Leeds

Institute of Medical Research, University of Leeds (EB and DF: awarded to SG, CM, AK, and RF), Medical Research Council grant G0700124 (SG), and Medical Research Council grants (MC_UU_12014/8 and MR/N017552/1) (AK).

## Additional information

### Funding

| Funder | Grant reference number | Author |
| --- | --- | --- |
| Medical Research Council | G0700124 | Matthew J Bentham<br>Stephen Griffin |
| University of Leeds | LIMR Studentship | Emma Brown<br>Daniella A Lefteri<br>Richard Foster<br>Antreas C Kalli<br>Stephen Griffin<br>Clive S McKimmie |
| Medical Research Council | MC_UU_12014/8 | Claire Donald<br>Alain Kohl |
| Medical Research Council | MR/N017552/1 | Claire Donald<br>Alain Kohl |
| Medical Research Council | MR/T016205/1 | Amy Moran<br>Stephen Griffin |
| UK Research and Innovation | Impact Acceleration Account (IAA) | Gemma Swinscoe<br>Stephen Griffin |

The funders had no role in study design, data collection, and interpretation, or the decision to submit the work for publication.

### Author contributions

Emma Brown, Formal analysis, Investigation, Visualization, Methodology, Writing – review and editing; Gemma Swinscoe, Methodology; Daniella A Lefteri, Investigation, Methodology, Writing – review and editing; Ravi Singh, Investigation, Visualization, Methodology, Writing – review and editing; Amy Moran, Hannah Beaumont, Matthew J Bentham, Investigation; Rebecca F Thompson, Data curation, Formal analysis, Supervision, Investigation, Visualization, Methodology; Daniel Maskell, Investigation, Methodology; Claire Donald, Resources, Methodology; Alain Kohl, Resources, Methodology, Writing – review and editing; Andrew Macdonald, Methodology, Writing – review and editing; Neil Ranson, Resources, Formal analysis, Methodology, Writing – review and editing; Richard Foster, Conceptualization, Supervision, Funding acquisition, Validation, Investigation, Visualization, Methodology, Writing – review and editing; Clive S McKimmie, Resources, Formal analysis, Supervision, Funding acquisition, Investigation, Methodology, Writing – review and editing; Antreas C Kalli, Data curation, Formal analysis, Supervision, Methodology, Writing – review and editing; Stephen Griffin, Conceptualization, Resources, Formal analysis, Supervision, Funding acquisition, Validation, Investigation, Visualization, Methodology, Writing - original draft, Writing – review and editing

### Author ORCIDs

Daniella A Lefteri https://orcid.org/0000-0002-9985-4254
Ravi Singh https://orcid.org/0000-0003-4344-4085
Claire Donald http://orcid.org/0000-0002-4370-0707
Antreas C Kalli https://orcid.org/0000-0001-7156-9403
Stephen Griffin https://orcid.org/0000-0002-7233-5243

### Ethics

Procedures were carried out in accordance with the United Kingdom Home Office regulations under the authority of the appropriate project and personal license (awarded to CSM and CSM/DL, respectively).

Decision letter and Author response
Decision letter https://doi.org/10.7554/eLife.68404.sa1
Author response https://doi.org/10.7554/eLife.68404.sa2

## Additional files

### Supplementary files
• MDAR checklist

### Data availability
All data generated or analysed during this study are included in the manuscript and supporting files; access to MD data and molecular models may be requested and, if accepted, accessed via MTA. Raw simulation data can be accessed via the Leeds Data Repository at the following DOI: https://doi.org/10.5518/1505.

The following dataset was generated:

| Author(s) | Year | Dataset title | Dataset URL | Database and Identifier |
|---|---|---|---|---|
| Brown ET, Kalli A, Griffin S | 2024 | Inhibitors of the Small Membrane (M) Protein Viroporin Prevent Zika Virus Infection - dataset | https://doi.org/10.5518/1505 | Leeds Data Repository, 10.5518/1505 |

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

## Appendix 1

Additional modelling solutions for oligomers with properties deemed less favourable compared to the compact hexameric complex lined by helix 3. This includes standard and protonated (His28$^{++}$) simulations for radial arrangements of the complex, complexes lined by helix 2 as well as heptameric assemblages across all conditions.

Each model and simulation is depicted as overall structure (**A**), cut-away and top–down images of complexes highlighting lumenal regions (neutral only, **B**), representative endpoint conformation following simulations (**C**), and HOLE profiles describing channel lumenal apertures for start (blue) and final (red, green, and black) conformations (**D**).

A

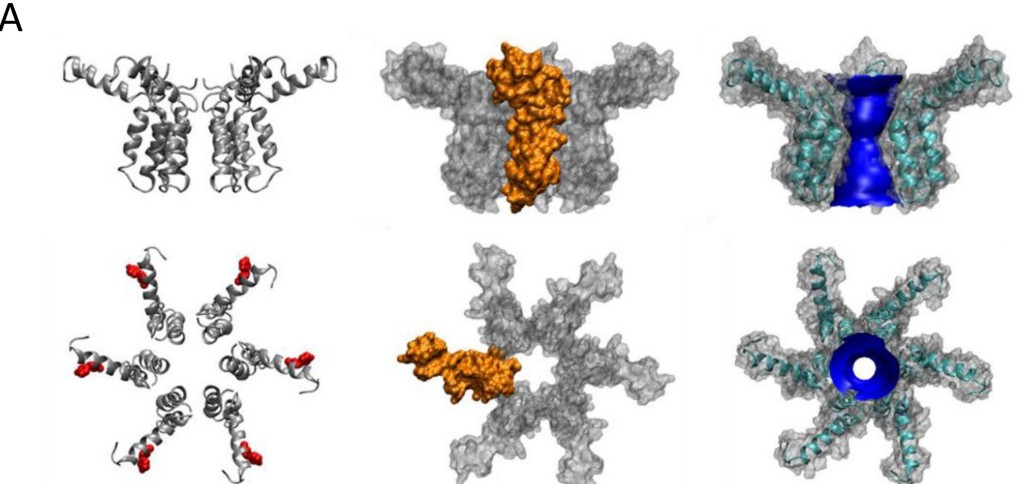

Radial lumenal helix 3, hexamer model, neutral 0 ns conformation

B

LEU68
ILE67
LEU64
TYR63
LYS60

C

200 ns

D

< 1.15 Å radius > 2.30 Å

z dimension (Å)

pore radius (Å)

LEU68
ILE67
LEU64

200 ns hole profile (neutral)

**Appendix 1—figure 1.** 'Radial' hexameric channel model with the lumen lined by helix 3. (**A**) Side (top) and plan (bottom) views of channel complexes as ribbon, space-fill (with protomer highlighted in gold), and ribbon/space-fill with HOLE lumenal profile. Structures energy minimised at neutral pH prior to simulation (i.e. 0 ns). His28 highlighted in red. (**B**) Cut-away (left) and top view of complex illustrating residues lining the lumen prior to simulation. (**C**) Examples of structures at 200 ns from four atomistic simulations, showing plan, side/cut-away, and top-ribbon views. (**D**) Hole profiles of the lumen following four separate atomistic simulations, with illustration of lumenal constriction.

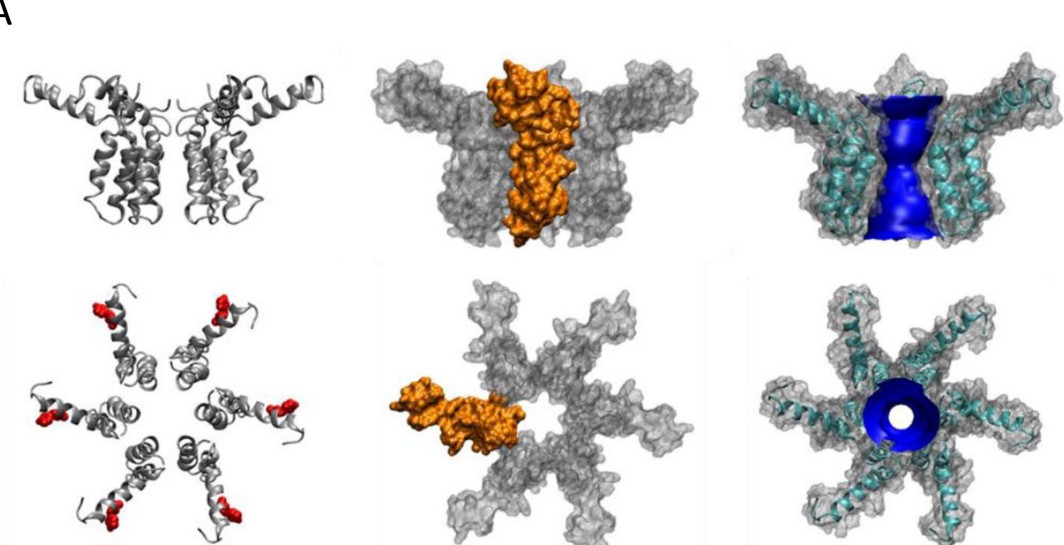

Radial lumenal helix 3, hexamer model, protonated 0 ns conformation

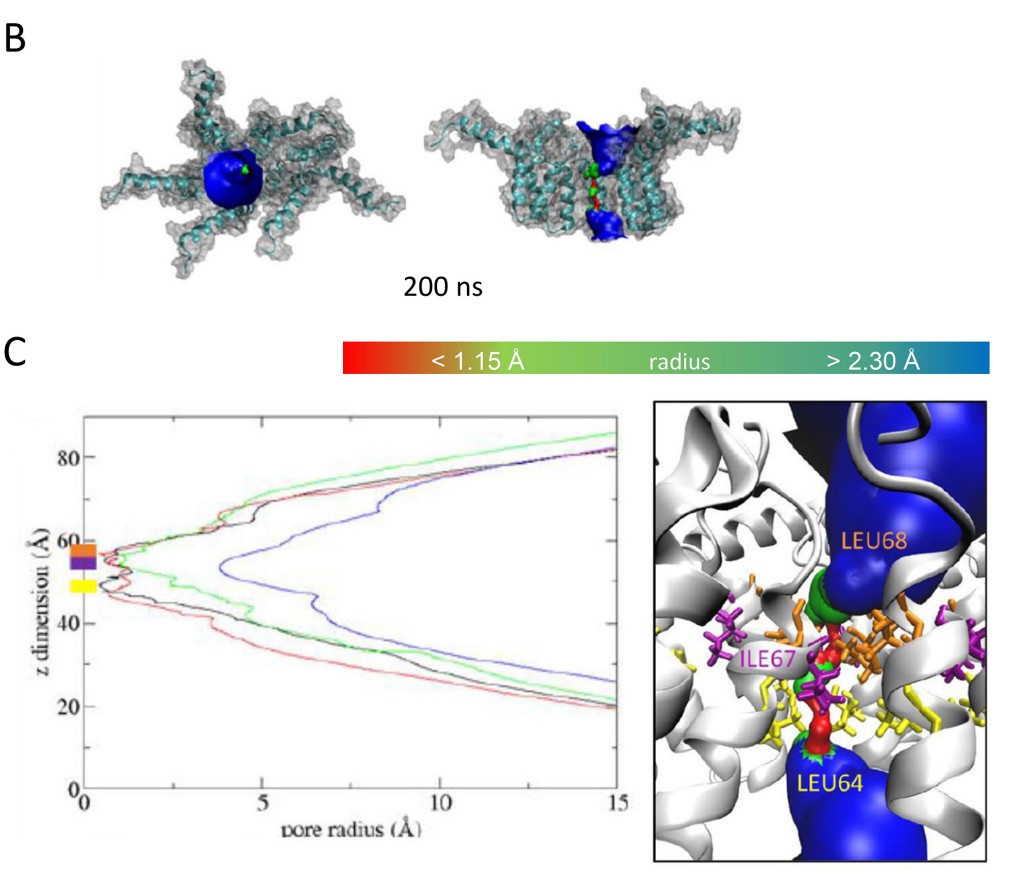

200 ns

200 ns hole profile (protonated)

**Appendix 1—figure 2.** As *Figure 1*, modelling acidic environment using dual protonation of the His28 residue.

A

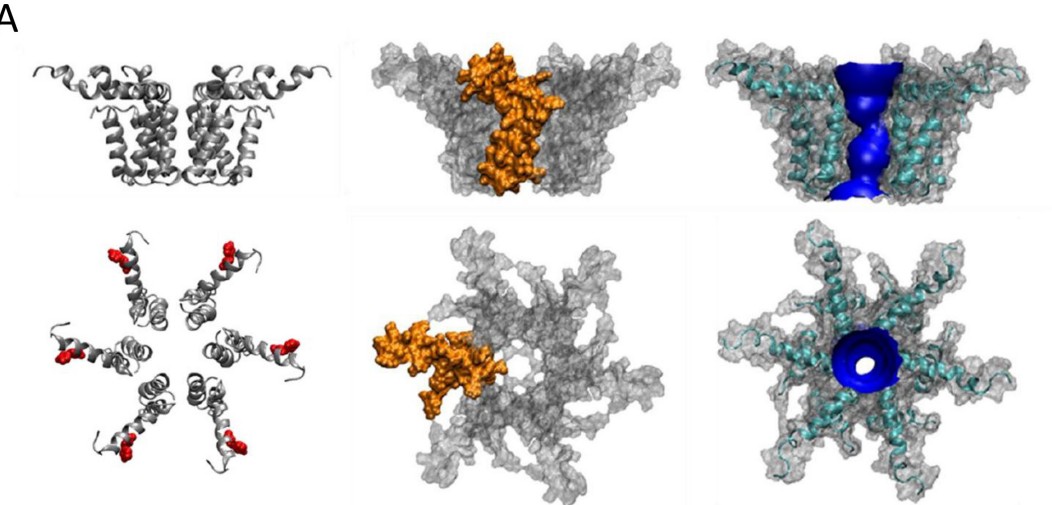

Radial lumenal helix 2, hexamer model, neutral 0 ns conformation

B

PRO40
LEU44
ALA47
TRP51

C

200 ns

D

< 1.15 Å          radius          > 2.30 Å

200 ns hole profile (neutral)

PRO40
ALA45
LEU44
TRP51

**Appendix 1—figure 3.** 'Radial' hexameric channel model with the lumen lined by helix 2. (**A–D**) as per *Figure 1*.

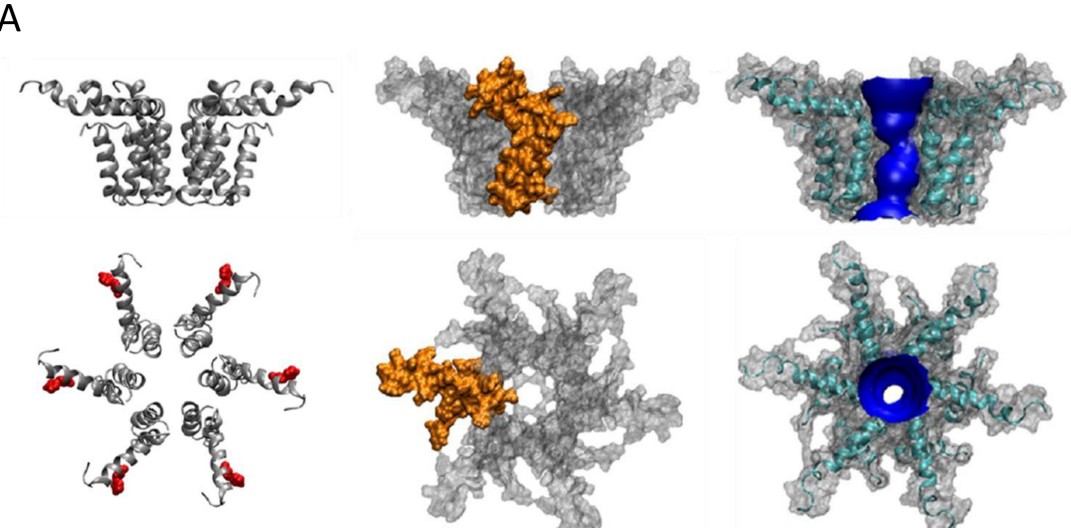

Radial lumenal helix 2, hexamer model, protonated 0 ns conformation

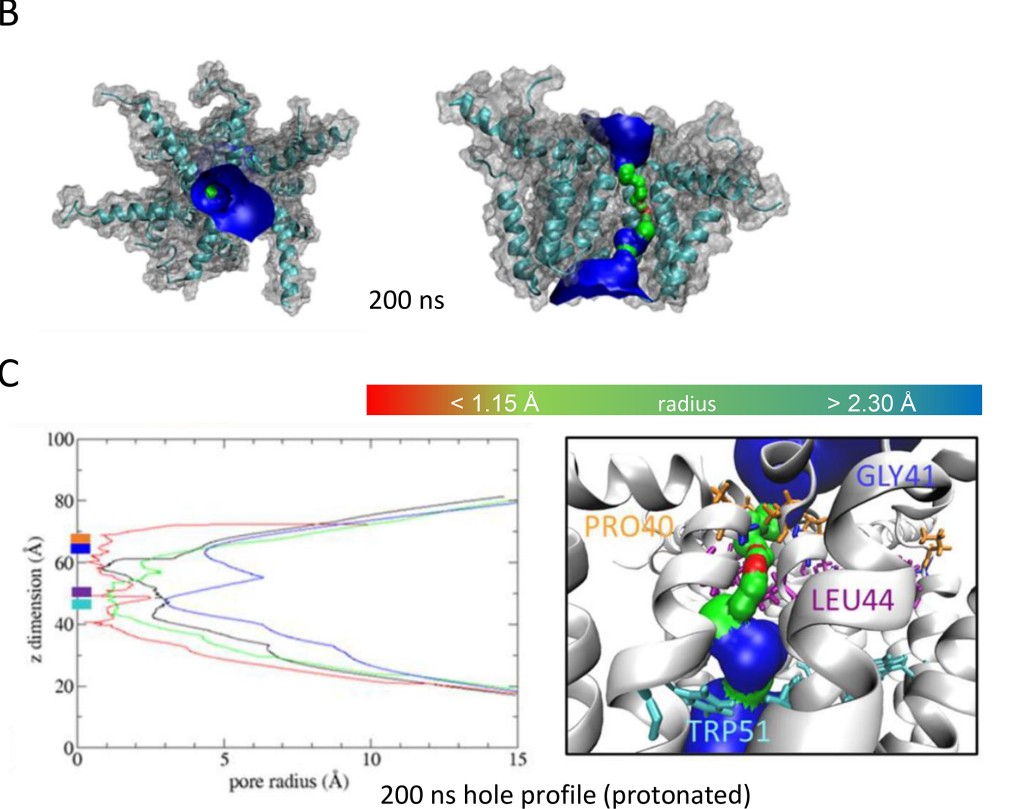

200 ns hole profile (protonated)

**Appendix 1—figure 4.** As *Figure 3*, modelling acidic environment using dual protonation of the His28 residue.

A

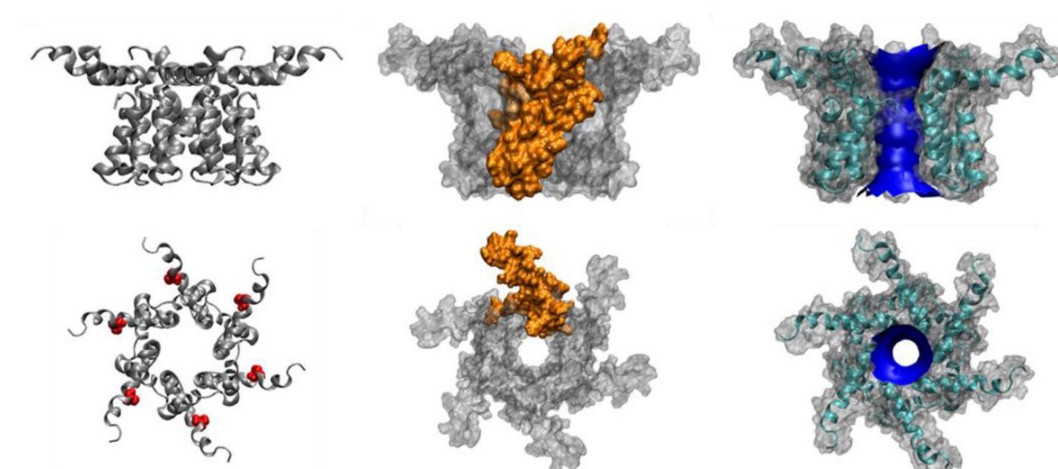

Compact lumenal helix 2, hexamer model, neutral 0 ns conformation

B

PHE42
ALA45
ILE49
LEU52

C

200 ns

D

< 1.15 Å          radius          > 2.30 Å

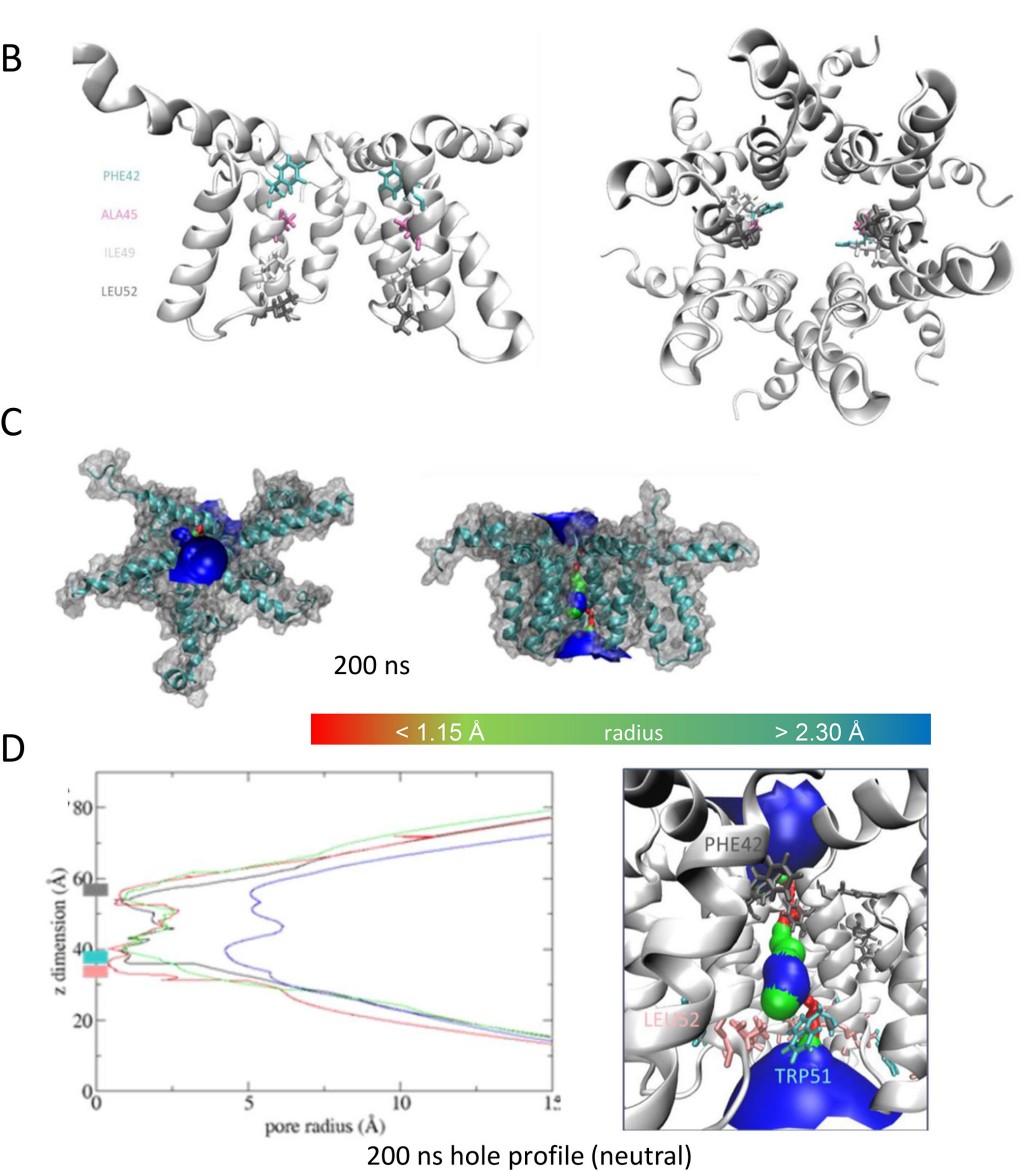

200 ns hole profile (neutral)

**Appendix 1—figure 5.** 'Compact' hexameric channel model with the lumen lined by helix 2. (**A–D**)as per *Figure 1*.

A

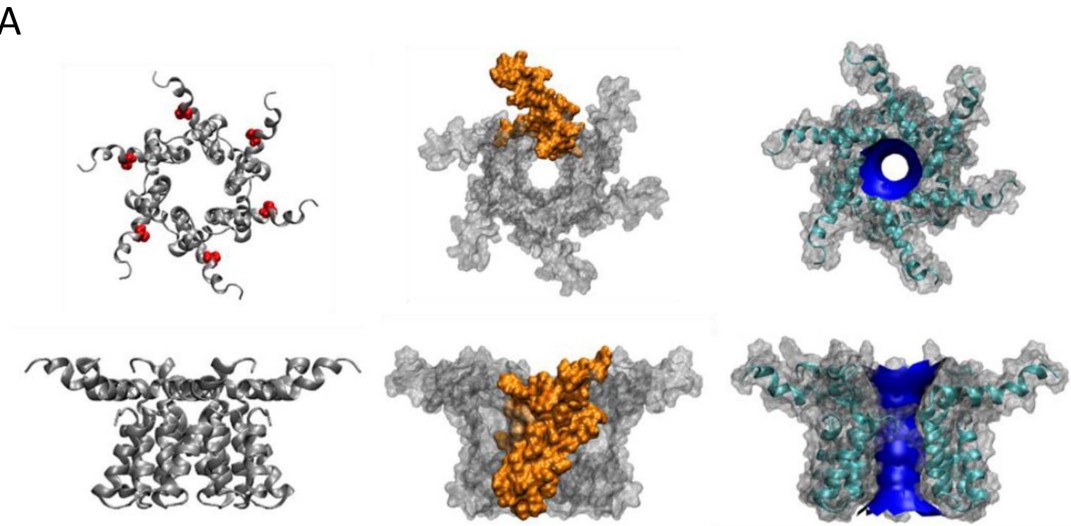

Compact lumenal helix 2, hexamer model, protonated 0 ns conformation

B

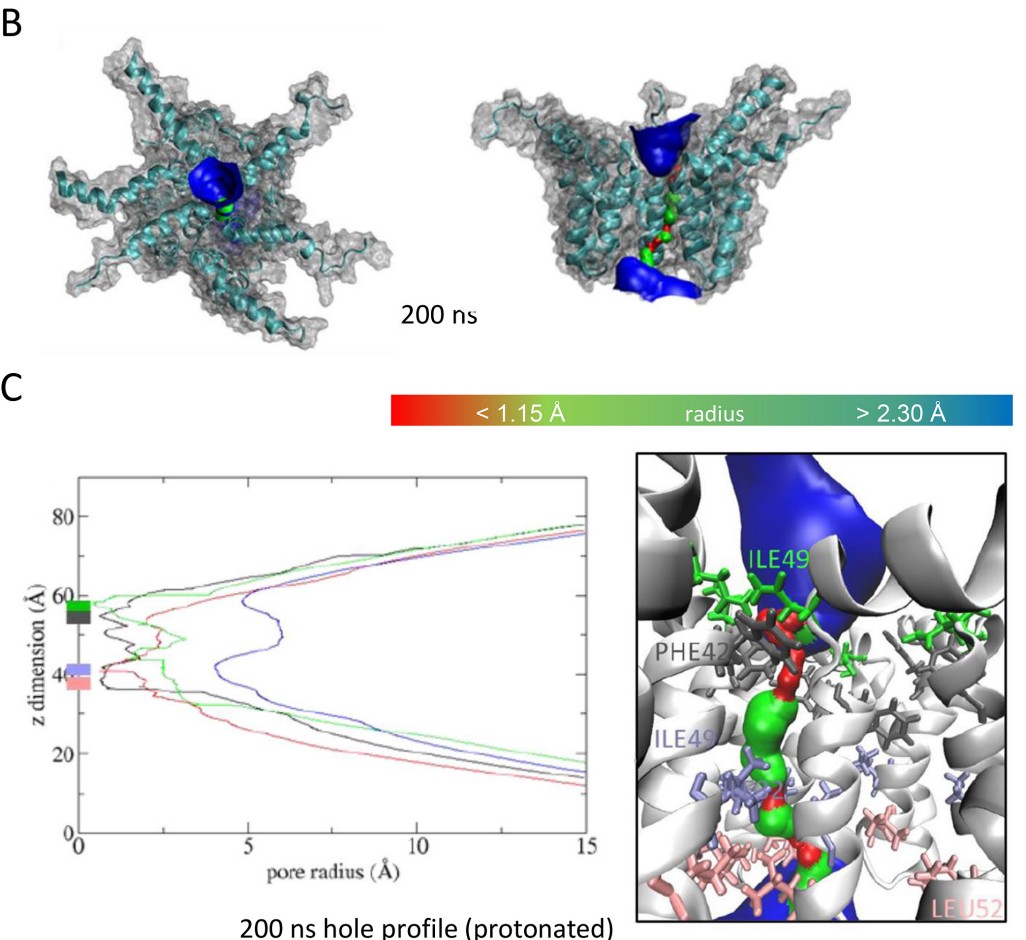

**Appendix 1—figure 6.** As *Figure 5*, modelling acidic environment using dual protonation of the His28 residue.

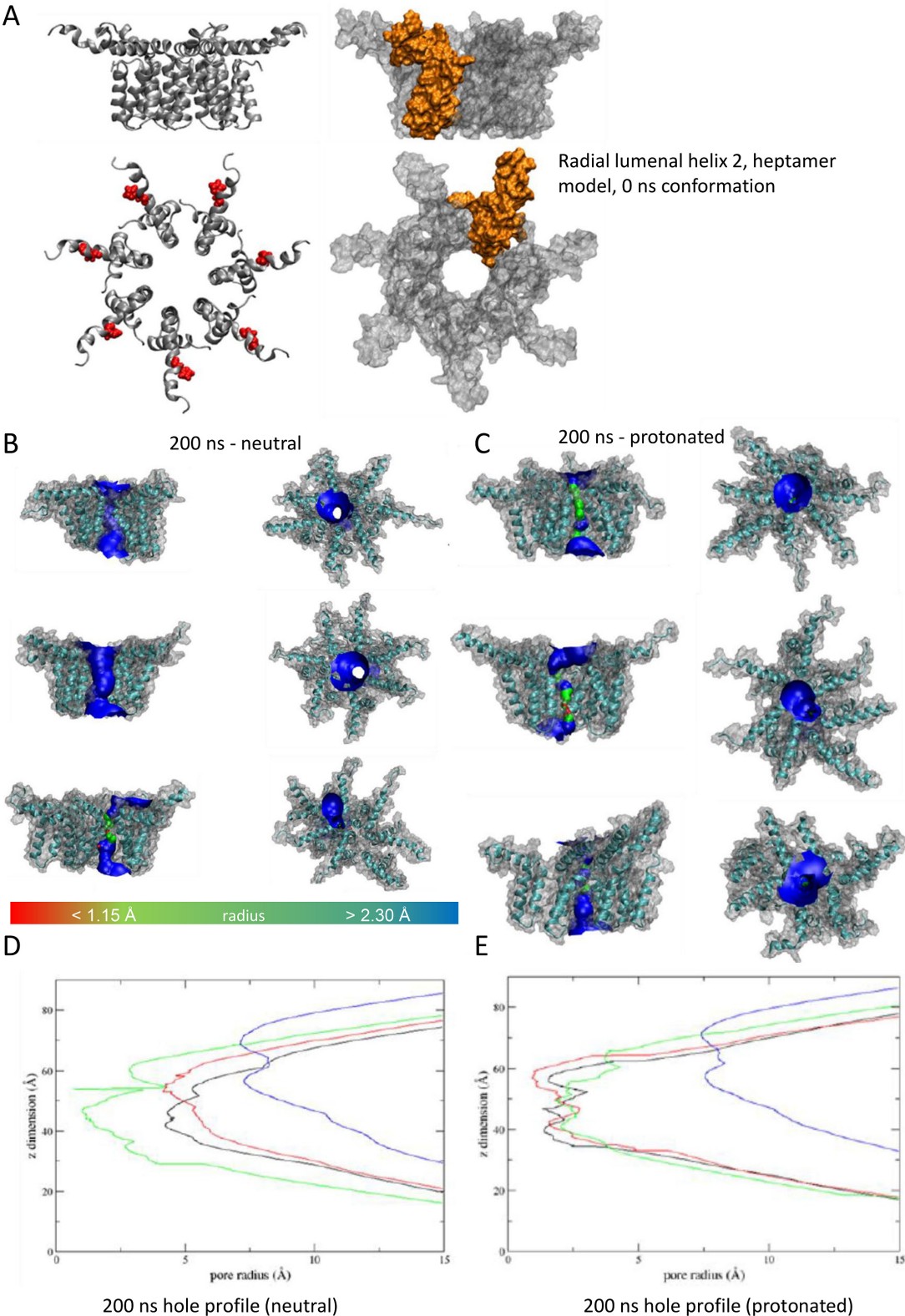

**Appendix 1—figure 7.** 'Radial' heptameric channel model with the lumen lined by helix 2. (**A**) Side (top) and plan (bottom) views of channel complexes as ribbon or space-fill (with protomer highlighted in gold). (**B**) Example conformers following 200 ns simulation at neutral pH, or (**C**) with double-protonated His28. (**D, E**)HOLE programme lumenal profiles corresponding to B and C.

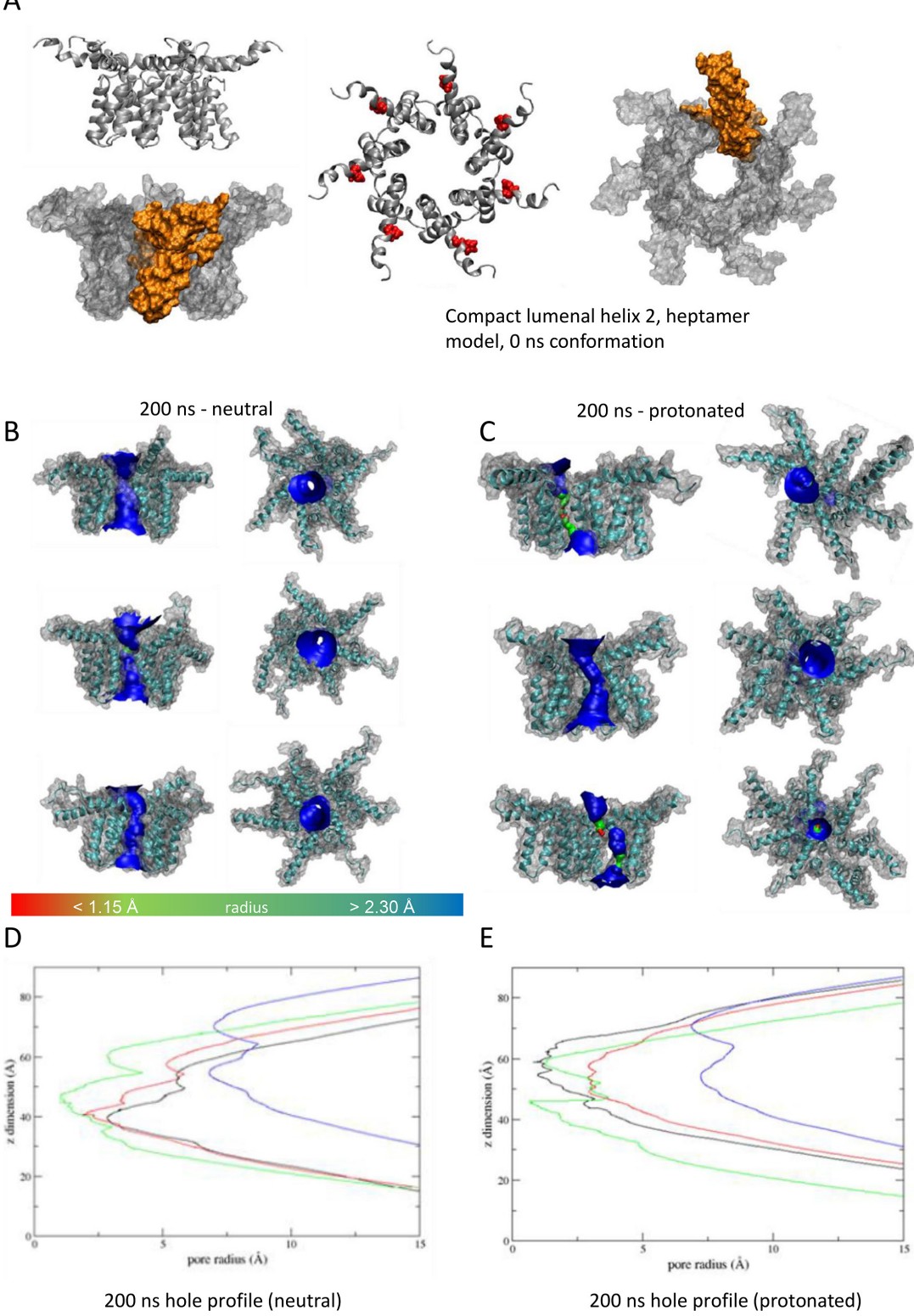

A

Compact lumenal helix 2, heptamer
model, 0 ns conformation

200 ns - neutral

B

C

200 ns - protonated

< 1.15 Å          radius          > 2.30 Å

D

E

200 ns hole profile (neutral)

200 ns hole profile (protonated)

**Appendix 1—figure 8.** 'Compact' heptameric channel model with the lumen lined by helix 2. (**A–E**) as per *Figure 7*.

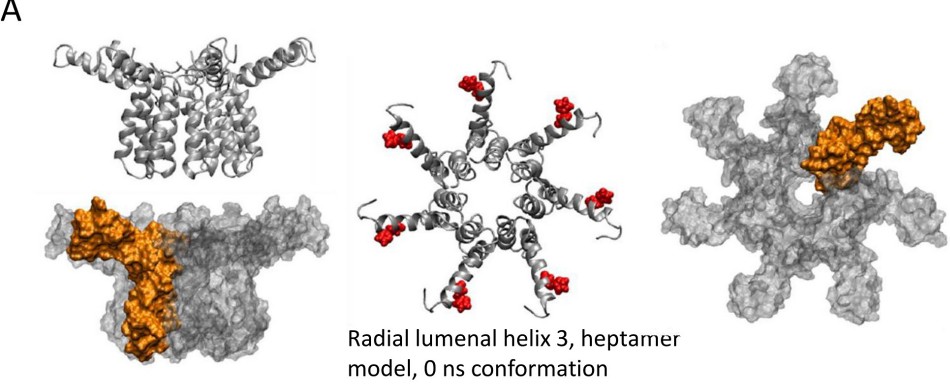

A

Radial lumenal helix 3, heptamer
model, 0 ns conformation

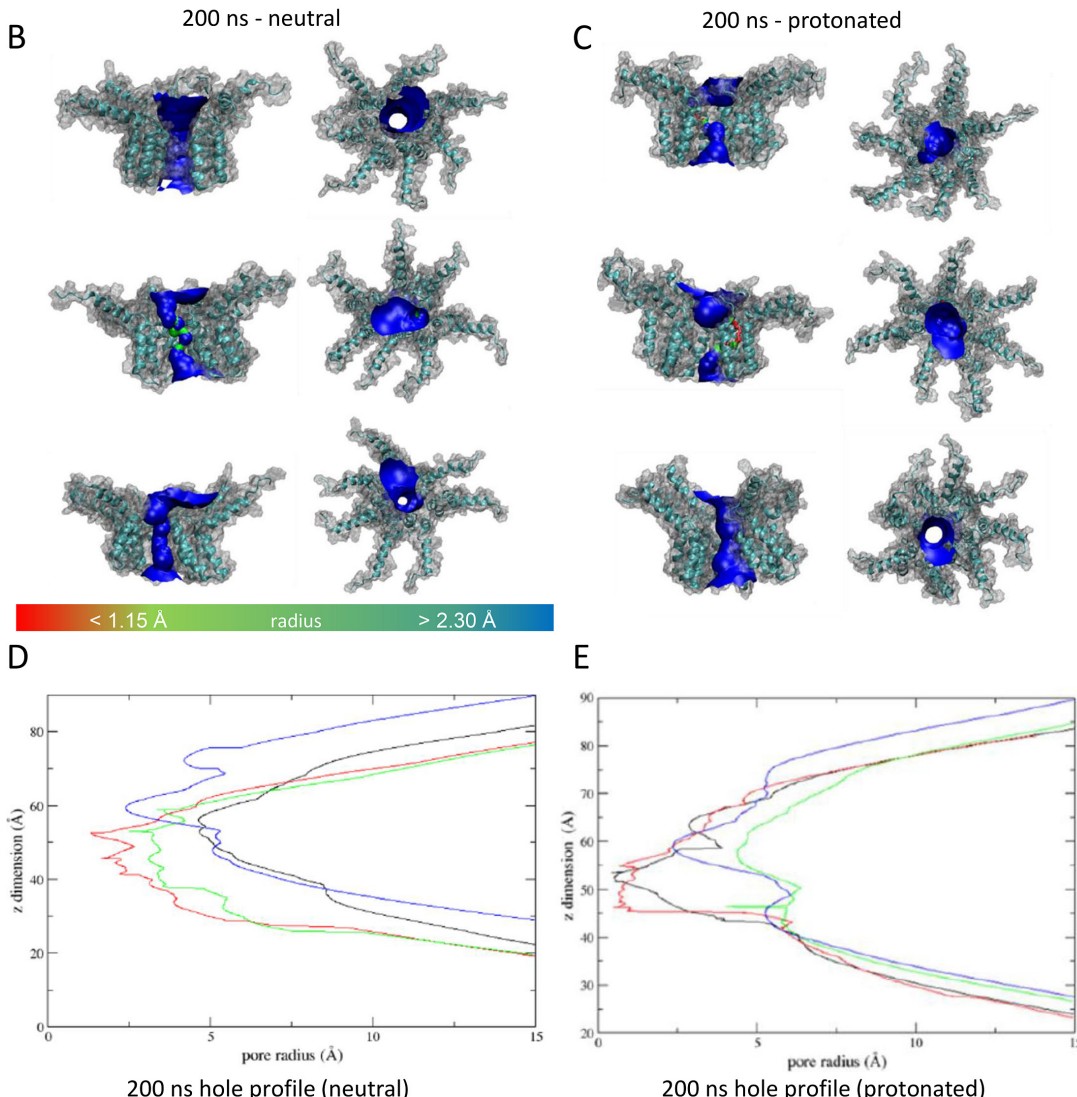

B    200 ns - neutral

C    200 ns - protonated

< 1.15 Å          radius          > 2.30 Å

D

200 ns hole profile (neutral)

E

200 ns hole profile (protonated)

**Appendix 1—figure 9.** 'Radial' heptameric channel model with the lumen lined by helix 3. (**A–E**) as per *Figure 7*.

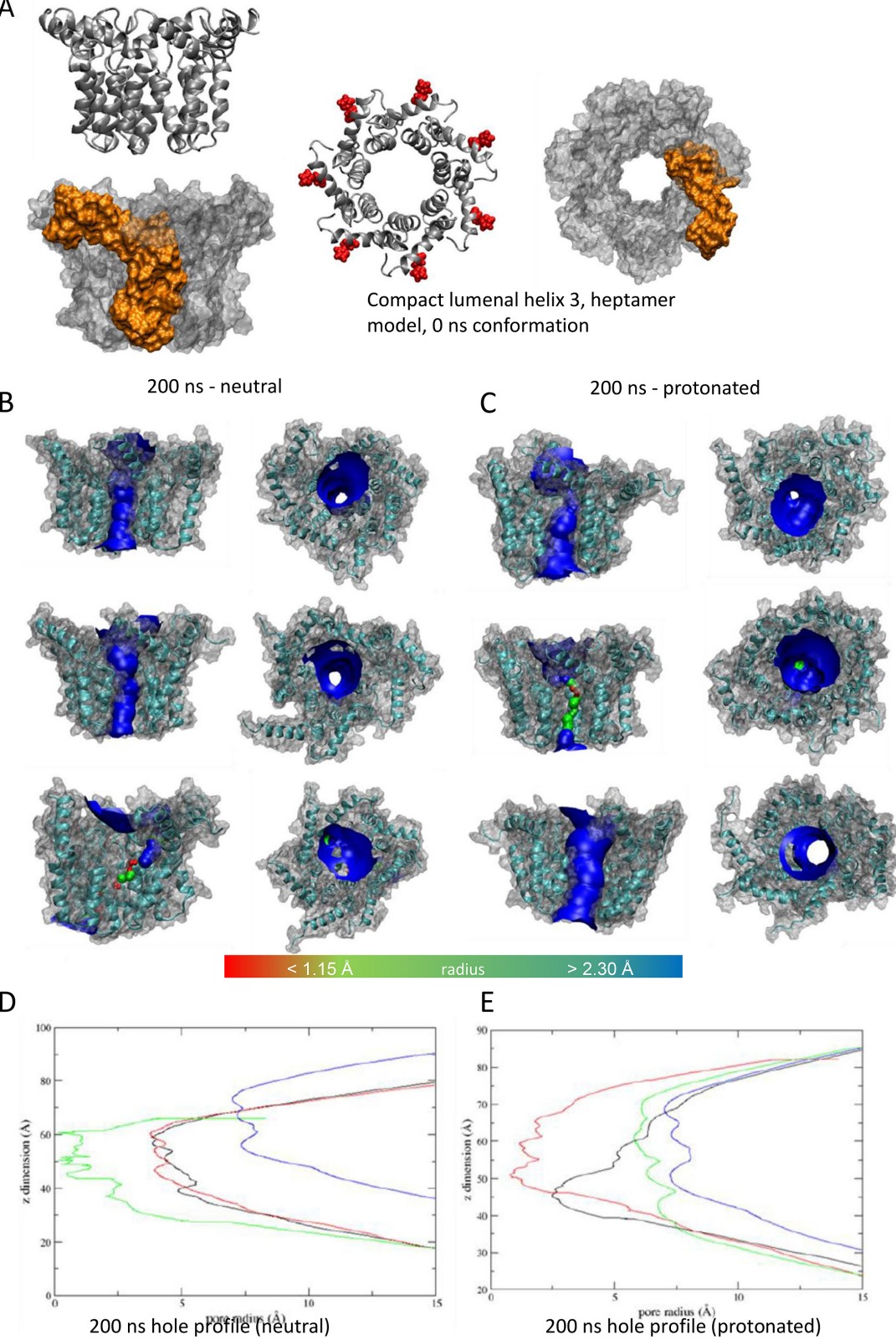

A

Compact lumenal helix 3, heptamer model, 0 ns conformation

200 ns - neutral

200 ns - protonated

B

C

< 1.15 Å    radius    > 2.30 Å

D

E

200 ns hole profile (neutral)

200 ns hole profile (protonated)

**Appendix 1—figure 10.** 'Compact' heptameric channel model with the lumen lined by helix 3. (**A–E**) as per *Figure 7*.

