## [Editor Report]

This study presents a valuable finding on the viroporin activity of the ZIKV M protein. The evidence supporting the claims of the authors is solid. The work will be of interest since M protein could be a relevant target for the development of new therapies.

---

## [Decision Letter]

**Decision letter after peer review:**

Thank you for submitting your article "Inhibitors of the Small Membrane (M) Protein Viroporin Prevent Zika Virus Infection" for consideration by *eLife*. Your article has been reviewed by 3 peer reviewers, including Nir Ben-Tal as the Reviewing Editor and Reviewer #1, and the evaluation has been overseen by Olga Boudker as the Senior Editor.

Essential revisions (for the authors):

Experiments:

1. Small molecule Inhibitors of ZIKV fusion targeting the E protein have been recently described (see Pitts et al. DOI:10.1016/j.antiviral.2019.02.008, or Li et al. DOI:10.1021/acsinfecdis.8b00322). Since similar inhibitors identified in this work appear to exert their action at the level of viral entry, they could also interfere with receptor binding, viral particle integrity or membrane fusion. The authors should add direct experimental evidence that the inhibitors indeed operate on the M protein. For example, in their previous work, where this issue was covered, the authors used Lentiviral vectors pseudotyped with viral glycoproteins to demonstrate the specificity of the inhibitors towards the viroporin target.

Related to this, to prove direct binding of the compounds to M it would be interesting to measure experimentally the process by ITC (detergent-stabilized samples?) or SPR (see for instance: https://doi.org/10.1016/j.bbamem.2015.12.028).

2. Permeability assays. Bilayers made of PC:PA might not reproduce adequately the viral membrane (thickness, potential interactions with specific lipids). Lipid compositions that approach more closely that of the ZIKV envelope should be used. In this regard, given the high curvature of the viral membrane (small radius of the particle), SUVs or LUVs produced by extrusion through 0.05 um-pore filters might be more adequate model systems to assay M-induced permeability than the larger vesicles used by the authors.

3. The authors claim that helix 1 and protonation of His28 therein are required for M channel function. In the channel model (Figure 6A), to span the bilayer, Helix 1 in a tilted angle combines with the following helix-turn-helix motif. It remained unclear if the authors acknowledge this key structural role of Helix 1. Are monomers of δ-Helix 1 spanning the bilayer in their simulations (Figure 7D)? The importance of helix 1 and His28 for M function can be easily tested in liposome permeabilization assays using synthetic δ-Helix 1 or peptides lacking His at position 28.

4. Figure 8D: rimantadine inhibition is rather low (at 1 μM and within this particular assay). Is it good enough to support the model? JK3/42 inhibition looks more convincing but it is surprising that there is no inhibitor with, say, 10% activity compared to DMSO. And anyway, we probably need a real dose response curve, rather than a single concentration.

5. Figure 9. Full dose response curves are required at least for the most promising hits.

Computations:

6. To prove that M protein is indeed a viroporin the authors should show that their in-silico models indeed translocate ions with specificity. A general methodology to show ion channel activity has been previously described in terms of the total flux of particles for p7 of the Hepatitis C virus (see Chandler et al. PLoS Comp. Bio. https://doi.org/10.1371/journal.pcbi.1002702).

7. The statistical treatment of the simulations needs to be further developed. Although, multiple replicates for each simulations are provided, there is no analysis that combines the results from the multiple replicates.

8. An assessment of the quality of the computationally derived models is lacking. The RMSD traces show values that are typical of intrinsically disordered proteins or poor structural integrity raising questions on the quality of the models and the ability to derive biologically relevant interpretations from the data.

9. The radius of the oligomeric model of the M protein was introduced as an ad-hoc parameter that was never optimized nor explored in detail. Simulations and analysis that prove the robustness of the results to the parameter should be performed. A similar issue is encountered in terms of the tilt-angle of the protein.

10. Details related to the relaxation of the lipid bilayer are not provided. Similarly, convergence of lipid-protein interfaces is not presented so it is impossible to assess if the simulations are well converged.

11. "Simulations of the two species within a POPC lipid bilayer revealed that the single helix rapidly began to revert to a hairpin-like structure (Figure 4c, d).": The figure does not support this statement. To judge this statement the reader needs to see a hairpin structure towards the end of the simulation and a plot of the RMSD of current conformation to the hairpin conformation from cryoEM as a function of simulation time.

12. "The N-terminal extra-membranous helix regulates formation of predicted M dimers": What is the point in this section? Are the dimers physiologically relevant? In which context?

13. "Molecular dynamics simulations favour the formation of compact hexameric channel complexes": The simulations started from tentative oligomeric structures, which is very risky because they may explore irrelevant regions of conformational space. Indeed, most of these tentative models collapsed and were rejected. However, there is no compelling evidence that the "compact hexameric channel complex" is physiologically relevant.

14. Rimantadine docking (Figure 8B): Without hydrogen bonds that provide specificity, the suggested docking pose does not look particularly good. How does this pose compare with that of binding to the influenza M2 channel, where we know that binding is real?

15. "Thus, inhibitor effects and docking data supported the presence of more than one distinct druggable binding site within M complexes, to exploit via novel chemical series": A bold statement that is not supported by any data. There is no direct evidence that links the predicted docking poses and inhibitory data. Especially that theoretically the compounds may act on a completely different target. Either another viral protein or a host protein that interfaces with the virus.

16. "Whilst it is possible that the canonical targets for these repurposed compounds might exert indirect antiviral activity rather than acting upon M, the chances of this occurring for the complete set were remote given their diversity": Incorrect statement. Theoretically each compound might affect its own unique target.

---

## [Author Response]

Essential revisions (for the authors):Experiments:1. Small molecule Inhibitors of ZIKV fusion targeting the E protein have been recently described (see Pitts et al. DOI:10.1016/j.antiviral.2019.02.008, or Li et al. DOI:10.1021/acsinfecdis.8b00322). Since similar inhibitors identified in this work appear to exert their action at the level of viral entry, they could also interfere with receptor binding, viral particle integrity or membrane fusion. The authors should add direct experimental evidence that the inhibitors indeed operate on the M protein. For example, in their previous work, where this issue was covered, the authors used Lentiviral vectors pseudotyped with viral glycoproteins to demonstrate the specificity of the inhibitors towards the viroporin target.Related to this, to prove direct binding of the compounds to M it would be interesting to measure experimentally the process by ITC (detergent-stabilized samples?) or SPR (see for instance: https://doi.org/10.1016/j.bbamem.2015.12.028).

This is an important point to address that we had not previously explored given that pseudotyping with Flavivirus E glycoproteins is dependent upon the chaperone function of the (pr)M protein. We are confident that rimantadine and the other drugs identified herein do not interfere with membrane integrity as there are no effects upon liposomes in the absence of M (or indeed other viroporins).

The receptor interactions undertaken by ZIKV are complicated due to interactions with multiple host factors as well as the existence of a spectrum of virion species. See for example, https://doi.org/10.1016/j.meegid.2019.01.018 and https://doi.org/10.1038/s41467-020-17638-y. Thus, determining effects upon specific E-receptor interactions is beyond the scope of this work.

However, in recognition of this important point, we investigated the integrity of ZIKV particles within iodixinol gradients (see new figure 7). First, consistent with the notion that ZIKV particles exist in different conformations, we witnessed two distinct peaks of infectivity, with a low-density fraction (~1.11 g/mL), only apparent in less diluted samples (1/10 dilution, see new supplementary figure 14). By contrast, the higher density fraction (1.13-1.14 g/mL) remained measurable upon sample dilution (1/100 or 1/1000) and contained the vast majority of particle infectivity. Infectivity within the higher density fraction was dramatically reduced by the addition of rimantadine, GNF 5837, or AA 29504, whereas minimal differences were observed in the lower density fraction. This was not due to virion lysis as the specific infectivity of particles in the peak fractions was diminished, normalised for genomic RNA content by RT-qPCR.

We acknowledge that biophysical characterisation of M-inhibitor interactions via SPR would be a desirable aim, but the establishment of the associated experimental pipeline is beyond the scope of this work, in particular due to the more hydrophobic nature of M compared with Vpu in the article mentioned, but also because of the difficulty assessing the predicted membrane-exposed binding site and allowing for partitioning of drug molecules into the bilayer. Assessments of the M2 peripheral binding site via such methods have, consequently, demonstrably favoured the interactions within the lumen compared to the periphery.

2. Permeability assays. Bilayers made of PC:PA might not reproduce adequately the viral membrane (thickness, potential interactions with specific lipids). Lipid compositions that approach more closely that of the ZIKV envelope should be used. In this regard, given the high curvature of the viral membrane (small radius of the particle), SUVs or LUVs produced by extrusion through 0.05 um-pore filters might be more adequate model systems to assay M-induced permeability than the larger vesicles used by the authors.

We use PC:PA bilayers as a model system because the design of the dye release assay requires that viroporins insert themselves spontaneously into the membranes. Over twenty years ago we established that use of acidic PA promoted this most efficiently, most likely due to regions of basic charge present in the majority of viroporins that appear to mediate this process. For example, our first major focus was the HCV p7 protein, which contains a highly conserved basic loop that we later showed was essential for membrane integration.

We have undertaken various MD simulations using membrane compositions based upon that published for the West Nile virus (WNV) membrane, the closest proxy we found at the time to the ZIKV envelope. We essentially saw no difference. However, the high cholesterol content of this and other RNA virus envelopes is not suited to dye release assays as more solid-phase membrane regions are refractory to viroporin insertion. Accordingly, such membranes give poor activity in dye release assays, and we also witnessed partitioning of HCV p7 (with a GST tag) into the fluidic regions of mixed membranes using atomic force microscopy. AFM data are unpublished as the project was not taken forwards, but we can gladly provide them if helpful.

We had not considered the use of smaller vesicles previously and were intrigued to follow this suggestion. Pleasingly, as shown in revised figure 2, an equivalent concentration of M peptides led to efficient and exponential dye release, comparable, albeit slightly lower than, standard assay conditions. We did not titrate the peptide under these conditions, and surmise that the altered molar ratios might explain the modest decrease in channel activity. We note that use of the smaller 0.05 mm filters in our extrusion setup (Avanti) is challenging experimentally, requiring considerable force, causing leaks, shearing membranes, and taking much longer to extrude. We therefore do not intend to use this routinely, but we agree it was an excellent suggestion.

3. The authors claim that helix 1 and protonation of His28 therein are required for M channel function. In the channel model (Figure 6A), to span the bilayer, Helix 1 in a tilted angle combines with the following helix-turn-helix motif. It remained unclear if the authors acknowledge this key structural role of Helix 1. Are monomers of δ-Helix 1 spanning the bilayer in their simulations (Figure 7D)? The importance of helix 1 and His28 for M function can be easily tested in liposome permeabilization assays using synthetic δ-Helix 1 or peptides lacking His at position 28.

It is right that we have not shown experimentally that helix 1 is required for channel function, although the simulation where it was lacking was compelling; Helix 1 does not span the membrane, but it lies parallel to the membrane in the lipid/water interface with some minor insertion in the bilayer. Only helices 2 and 3 span the membrane. We can only speculate of the role of Helix 1 but it can be: 1) it stabilizes the channels via its interactions with the bilayer surface (like an anchoring point) or 2) by contributing to the interactions with adjacent subunits. Indeed, in the simulation without helix 1, or in the “radial” hexamer simulations in which helix 1 does not interact with the other subunits, the channel is unstable. Hence, it can be either scenario, or a combination of the two.

The suggestion to generate a His28Ala peptide was taken forward and we tested this in the pH responsive dye release assay alongside the wild-type peptide. Surprisingly, this peptide retained virtually identical characteristics and increased activity at more acidic pH.

We agree that whilst our simulations do show some differences between the protonated and aprotonated channels, they are relatively small. This may be the reason that we don’t detect differences in the His28Ala experiment. We originally chose His28 as it is the most obvious site for becoming protonated at physiological pH. However, as the reviewer also notes, other amino acids are present within the M sequence that could become protonated. Thus, it may be that modelling the His28++ protein served to emulate other protonation events, rather than representing the precise biochemical mechanism; the pKa of amino acids within proteins can be significantly influenced by their local environment. Protonation could also feasibly promote a localised imbalance of acidic and basic charges, involving any of several Thr, Tyr, Asn, Arg, or Lys residues in the sequence.

We have amended the relevant section of the results and other sections and included additional discussion considering these new findings.

4. Figure 8D: rimantadine inhibition is rather low (at 1 μM and within this particular assay). Is it good enough to support the model? JK3/42 inhibition looks more convincing but it is surprising that there is no inhibitor with, say, 10% activity compared to DMSO. And anyway, we probably need a real dose response curve, rather than a single concentration.

We fully accept that rimantadine potency is relatively low in vitro; this is expected as it is a generic broad-specificity channel blocker. We have attempted dose-response curves for most of the main hits, as well as rimantadine. These are, frustratingly, complicated by an assay artefact that occurs for some, but by no means all viroporin/inhibitor combinations; for example, M2 shows hyperactivation with HMA in our hands, and several of our JK series have the same effect upon p7 despite potently lowering activity when used at lower concentrations. This is not an effect of the drug upon lipids as it does not occur in the absence of viroporins. Other examples of compound hyperactivation are present in the drug screening data (Figure S10), including GBR12909 and abacavir.

Rimantadine is used in lieu of more tailored compounds to identify druggable targets because of its inherent promiscuity. The JK series were used to build an SAR for the peripheral binding site in p7 channel complexes (also a structure-guided channel model), which resembles the cavity in the predicted M complex. There was every chance that this series may not in fact work, but given that two did, we presume the peripheral site is a druggable option, supported by the docking and improved efficacy of our screening compounds. As such, we have moved the JK data to supplementary alongside adamantane analogues, so as not to disrupt the flow of the piece.

5. Figure 9. Full dose response curves are required at least for the most promising hits.

We agree, and have conducted 8-point EC50 curves for the effects of GNF 5837 and AA 29504, yielding values of 748.1 nM and 11.37 mM, respectively – see revised figure 6F, G.

Computations:6. To prove that M protein is indeed a viroporin the authors should show that their in-silico models indeed translocate ions with specificity. A general methodology to show ion channel activity has been previously described in terms of the total flux of particles for p7 of the Hepatitis C virus (see Chandler et al. PLoS Comp. Bio. https://doi.org/10.1371/journal.pcbi.1002702).

We have checked our simulations, which do contain water columns prior to closing, for ion permeation. Given their relatively short timescales, this was deemed unlikely, nevertheless we see water going through the channel which show a first indication that this can potentially form a channel, now shown in revised figure 5c.

We agree that exploring this is important. However, to study ion permeation and specificity is a major set of simulation work, perhaps a whole new study. Indeed, in the case of the article above, this comprises a self-contained study.

7. The statistical treatment of the simulations needs to be further developed. Although, multiple replicates for each simulations are provided, there is no analysis that combines the results from the multiple replicates.

We agree and we now show the RMSD analysis in Figure 4 as the average with standard deviation. For the simulation with the monomer and the hexameric structure which are the main simulations we discuss in the main text we have also added additional SI figures in which we show analyses that combine data from the simulation ensembles e.g., protein/lipid analysis.

8. An assessment of the quality of the computationally derived models is lacking. The RMSD traces show values that are typical of intrinsically disordered proteins or poor structural integrity raising questions on the quality of the models and the ability to derive biologically relevant interpretations from the data.

We agree and we now show the RMSD analysis in Figure 4 as the average with standard deviation. For the simulation with the monomer and the hexameric structure which are the main simulations we discuss in the main text we have also added additional SI figures in which we show analyses that combine data from the simulation ensembles e.g., protein/lipid analysis.

9. The radius of the oligomeric model of the M protein was introduced as an ad-hoc parameter that was never optimized nor explored in detail. Simulations and analysis that prove the robustness of the results to the parameter should be performed. A similar issue is encountered in terms of the tilt-angle of the protein.

The pore radius was not a fixed parameter, but it was a result of the complex assembly. We have made efforts to vary the model by using two different orientations of the monomer that are sterically feasible. We have also constructed pentamers, hexamers and heptamers that differ in their pore radius. Regarding the tilt angle again we used two different protein orientations. We could have tried more tilt angles, but one advantage of MD is that proteins have the option to optimise packing and tilt angles during the simulation.

10. Details related to the relaxation of the lipid bilayer are not provided. Similarly, convergence of lipid-protein interfaces is not presented so it is impossible to assess if the simulations are well converged.

We have now included calculated bilayer parameters (density profiles also the bilayer normal of the lipids and of the proteins and the simulation box X, Y, Z component for the atomistic simulation with the hexameric structure). We have also calculated and protein-lipid interfaces for the simulation with the monomeric and hexameric structures. These analyses show that the bilayer is properly functioning during the simulations. We also show convergence analysis (for 1us, 2us and 3us) of the protein/lipid interfaces for the monomeric simulations. This analysis shows that after 1us the coarse-grained simulation are converged.

11. "Simulations of the two species within a POPC lipid bilayer revealed that the single helix rapidly began to revert to a hairpin-like structure (Figure 4c, d).": The figure does not support this statement. To judge this statement the reader needs to see a hairpin structure towards the end of the simulation and a plot of the RMSD of current conformation to the hairpin conformation from cryoEM as a function of simulation time.

The reviewer is correct, we do not actually see a hairpin structure per se, i.e., one similar to the EM structure. However, we do see the tendency of the second helix to enter the bilayer. We already refer to this as a “hairpin-like” structure, but we have reworded this further.

12. "The N-terminal extra-membranous helix regulates formation of predicted M dimers": What is the point in this section? Are the dimers physiologically relevant? In which context?

As aforementioned, on reflection we agree that the inclusion of the dimerisation data is not essential to the manuscript, and this has now been removed.

13. "Molecular dynamics simulations favour the formation of compact hexameric channel complexes": The simulations started from tentative oligomeric structures, which is very risky because they may explore irrelevant regions of conformational space. Indeed, most of these tentative models collapsed and were rejected. However, there is no compelling evidence that the "compact hexameric channel complex" is physiologically relevant.

This approach has been used in other published studies, including that discussed above in relation to ion permeation (https://journals.plos.org/ploscompbiol/article?id=10.1371/journal.pcbi.1002702). It is not possible to assemble the hexameric structures by starting with six monomers in the bilayer because of the computational cost. We could have potentially used coarse-grained simulations but forming a hexamer starting from monomers may not be possible due to the lower resolution of these simulations. Therefore, we had to start with pre-assembled dimers. We also compared two helical orientations (i.e. with either helix 2 or 3 lining the lumen), each as pentamers, hexamers, and heptamers. Thus, we did in fact test different conformations that are in a different part of the energy landscape. Also, as the reviewer states, if the hexamers weren’t energetically stable, they would have collapsed during the simulations as did the others, including where helix 1 was removed.

In addition to the improved inherent stability of the hexameric model, it should be noted that Flavivirus particles comprise a regulated protein stoichiometry pertaining to the E and (pr)M proteins, which is always a multiple of both two and three in relation to mature and immature virion membrane protein arrangements. Hence, whilst it is theoretically feasible for mixed particles to assemble dimers and trimers in different combinations, if the mature virions predominate the most likely channel building blocks will be M dimers.

We also point out that we consider the hexamer model to be very much a tool by which to investigate the potential role and relevance of M protein channels. The first way in which the relevance of this model was tested was to use it as a template to derive candidate small molecule inhibitors. Whilst early-stage hits may or may not eventually be found to interact at the specific site against which they are targeted (to be confirmed via later investigations), the accumulation and enrichment of effective molecules compared to that expected by random chance is, in our opinion, supportive that the model is approaching biological relevance. Moreover, the correlation between predicted binding, in vitro and cell culture efficacy adds to this evidence base.

14. Rimantadine docking (Figure 8B): Without hydrogen bonds that provide specificity, the suggested docking pose does not look particularly good. How does this pose compare with that of binding to the influenza M2 channel, where we know that binding is real?

We agree that H-bonds will increase avidity and specificity for binding sites in general. However, the binding and orientation of rimantadine/amantadine even within M2 varies considerably between different structures. Even when appropriately oriented in the M2 lumen, the amine group interacts randomly with any one of the four His37 residues comprising the proton sensor within the channel. Moreover, binding at the peripheral, membrane-exposed binding site can also vary depending upon the environment in which peptides containing the peripheral site are analysed. Ultimately, neither rimantadine nor amantadine possess qualities that favour highly specific interactions. Instead, the compact adamantyl cage undergoes promiscuous interactions with multiple binding sites, which may or may not be augmented via H-bonding via the amine. Hence, we use this as a guide towards druggable sites, rather than any sort of benchmark, as described above, and this has led to some success including for rimantadine-resistant influenza (Scott *et al.*, PLoS Pathogens, 2020).

15. "Thus, inhibitor effects and docking data supported the presence of more than one distinct druggable binding site within M complexes, to exploit via novel chemical series": A bold statement that is not supported by any data. There is no direct evidence that links the predicted docking poses and inhibitory data. Especially that theoretically the compounds may act on a completely different target. Either another viral protein or a host protein that interfaces with the virus.

We agree and have amended the relevant section in the text. However, we note that tropomyosin receptor kinases, the target of GNF5837, are primarily expressed in the CNS and the thyroid with only very low expression in kidney tissues according to the human protein atlas. Moreover, we could find no reference to TRK expression in either Vero, or BHK cells. Similarly, GABAA receptors, the target of AA 29504, are not expressed in the kidney.

16. "Whilst it is possible that the canonical targets for these repurposed compounds might exert indirect antiviral activity rather than acting upon M, the chances of this occurring for the complete set were remote given their diversity": Incorrect statement. Theoretically each compound might affect its own unique target.

As above, we have amended the relevant text with focus upon target expression, although we cannot categorically rule out off-target effects as the reviewer points out.